# Efficient Credal Prediction through Decalibration

**Paul Hofman**[1,2,*], **Timo Löhr**[1,2,*], **Maximilian Muschalik**[1,2], **Yusuf Sale**[1,2],
**Eyke Hüllermeier**[1,2,3]
[1]LMU Munich, [2]Munich Center for Machine Learning (MCML), [3]DFKI (DSA)
[*]Equal Contribution
{paul.hofman, timo.loehr, maximilian.muschalik, yusuf.sale, eyke}@ifi.lmu.de

## Abstract

A reliable representation of uncertainty is essential for the application of modern machine learning methods in safety-critical settings. In this regard, the use of credal sets (i.e., convex sets of probability distributions) has recently been proposed as a suitable approach to representing epistemic uncertainty. However, as with other approaches to epistemic uncertainty, training credal predictors is computationally complex and usually involves (re-)training an ensemble of models. The resulting computational complexity prevents their adoption for complex models such as foundation models and multi-modal systems. To address this problem, we propose an efficient method for credal prediction that is grounded in the notion of relative likelihood and inspired by techniques for the calibration of probabilistic classifiers. For each class label, our method predicts a range of plausible probabilities in the form of an interval. To produce the lower and upper bounds of these intervals, we propose a technique that we refer to as decalibration. Extensive experiments show that our method yields credal sets with strong performance across diverse tasks, including coverage–efficiency evaluation, out-of-distribution detection, and in-context learning. Notably, we demonstrate credal prediction on models such as TabPFN and CLIP—architectures for which the construction of credal sets was previously infeasible.

## 1 Introduction

Modern machine learning (ML) is increasingly deployed in domains where decisions carry real consequences, from energy systems (Miele et al., 2023) and weather forecasting (Bülte et al., 2025) to healthcare (Löhr et al., 2024). In such domains, we need models that not only make accurate predictions, but also express what they do *not* know. A useful starting point is the distinction between *aleatoric* and *epistemic* uncertainty (Hüllermeier & Waegeman, 2021). Aleatoric uncertainty reflects irreducible randomness in the data. Epistemic uncertainty reflects limited knowledge and, in principle, can be reduced with more or better information. While standard probabilistic predictors capture the former, representing the latter typically requires higher-order formalism.

Credal sets, i.e., (convex) sets of probability distributions, offer such a view. Instead of committing to a single predictive distribution, a credal predictor returns a set of plausible distributions, thereby making epistemic uncertainty explicit (Levi, 1978; Walley, 1991). Credal methods have appealing semantics but can be computationally demanding: many pipelines rely on ensembles or approximate posteriors to explore the space of plausible models, which is difficult to justify for large and complex models such as foundation models, CLIP (Radford et al., 2021) or TabPFN (Hollmann et al., 2022).

We take a different route. Building on a likelihood-based notion of plausibility (Löhr et al., 2025), we construct credal predictions *from a single trained model* by *decalibration*: we systematically perturb the model's logits so that the resulting probabilities move away from the maximum-likelihood fit while staying within a prescribed relative-likelihood budget. For each class, this procedure yields a plausible probability interval; their product forms a credal set that reflects epistemic uncertainty

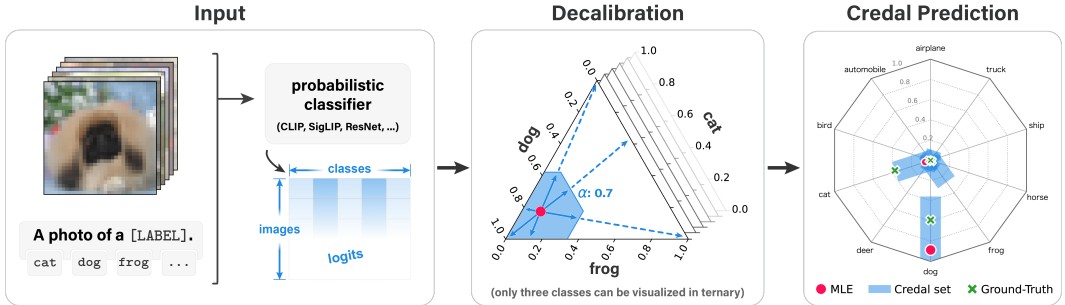

**Figure 1: Overview of Efficient Credal Prediction through Decalibration.** Given a probabilistic classifier (maximum likelihood estimate), our method *decalibrates* the predicted distributions by their logits. The resulting credal set contains the ground-truth distribution, as visualized in the *credal spider plot* (see Appendix C for an explanation). Note that we only show the decalibration of three classes for visualization purposes—in practice, all classes are decalibrated.

without retraining (cf. Figure 1). Intuitively, calibration adjusts probabilities to be more correct, whereas decalibration explores how far they can be pushed and still remain supported by the data.

In this light, our **contributions** are as follows. ①A model-agnostic, post-hoc method for credal prediction via *decalibration*, logit perturbations that produce class-wise plausible probability intervals under a relative-likelihood budget, yielding credal sets with the clear semantics "reachable without sacrificing more than a chosen fraction of training likelihood." The procedure requires *no retraining* and only logits, enabling use with large pretrained models. ②Theoretically, we show the relative-likelihood feasibility set induced by logit shifts is convex (and compact on an identifiability hyperplane); that upper class-wise bounds arise from a single convex optimization; and that in a one-dimensional, class-specific shift the plausible interval endpoints solve two convex programs with monotone probabilities, implying nested credal sets as the likelihood budget tightens. ③Empirically, across benchmarks our credal sets achieve favorable coverage–efficiency trade-offs and competitive out-of-distribution detection while reducing computational cost by orders of magnitude. The method enables credal prediction for previously out-of-reach models such as `TabPFN` and `CLIP`, and we introduce *credal spider plots* to visualize interval-based sets beyond three classes.

## 2  CREDAL PREDICTION BASED ON PLAUSIBLE INTERVALS

We assume a supervised classification setting, where $\mathcal{X}$ denotes the instance space, and $\mathcal{Y} = \{1, \ldots, K\}$ the finite set of class labels. Further, we assume that the learner has access to (i.i.d.) training data $\mathcal{D}_{\text{train}} = \{(\boldsymbol{x}^{(n)}, y^{(n)})\}_{n=1}^{N} \subset \mathcal{X} \times \Delta_K$. In this paper, we consider a hypothesis space $\mathcal{H}$ with hypotheses of the form $h : \mathcal{X} \to \Delta_K$, mapping instances $\boldsymbol{x} \in \mathcal{X}$ to probability distributions over $\mathcal{Y}$; by $\Delta_K$ we denote the set of all probability distributions on the label space $\mathcal{Y}$. Our primary concern is *predictive uncertainty*, i.e., the uncertainty about the predicted label $\hat{y}_q$ at a query point $\boldsymbol{x}_q \in \mathcal{X}$. While probabilistic predictors $\mathcal{X} \to \Delta_K$, $\boldsymbol{x}_q \mapsto h(\boldsymbol{x}_q) = p(\cdot \mid \boldsymbol{x}_q, h)$ do account for aleatoric uncertainty, they do not represent *epistemic* uncertainty about the predicted probability $p(\cdot \mid \boldsymbol{x}_q, h)$. To make this uncertainty explicit, we move from point predictions in $\Delta_K$ to sets and consider an uncertainty-aware, set-valued predictor $H : \mathcal{X} \to \mathcal{K}(\Delta_K)$, where $\mathcal{K}(\Delta_K)$ denotes a suitable family of subsets of the simplex (e.g., nonempty closed convex sets). In this view, the prediction at $\boldsymbol{x}_q$ is no longer a single vector $h(\boldsymbol{x}_q) \in \Delta_K$, but a set $H(\boldsymbol{x}_q) = \mathcal{Q}_{\boldsymbol{x}_q} \subseteq \Delta_K$, which we refer to as a *credal prediction*.

Credal sets have emerged as a compelling representation of (epistemic) uncertainty in contemporary machine learning research, yet there is no consensus on how to *construct* them in a principled and scalable way. We aim for a construction that is ($i$) statistically well-founded, ($ii$) semantically transparent, and ($iii$) computationally feasible for modern large models. Many existing pipelines either rely on Bayesian posteriors (Caprio et al., 2024a), thus inheriting prior sensitivity and computational burden, or on ad-hoc ensembling and heuristics that offer weak interpretability (Wang et al., 2025a). In contrast, following Löhr et al. (2025), we adopt a *likelihood*-based notion of plausibility that is

prior-free, data-driven, and well established in statistical inference. Concretely, relative (normalized) likelihood provides a scalar measure of model plausibility: a model is considered plausible at level $\alpha \in (0, 1]$ if its likelihood is at least an $\alpha$-fraction of the maximum likelihood in the model class. This yields an $\alpha$-indexed family of *plausibility regions* in parameter space, whose images in prediction space induce (class-wise) *plausible probability intervals*. These intervals serve as the *inputs* that generate a credal set in the simplex.

With respect to our desiderata $(i)$–$(iii)$, Löhr et al. (2025) already address $(i)$ and $(ii)$: the likelihood ratio supplies a prior-free, data-driven evidential scale that is standard in statistics, and normalizing by the maximum likelihood yields nested, interpretable $\alpha$-cuts (Antonucci et al., 2012). This viewpoint is well established, likelihood ratios underpin classical confidence regions and tests and admit a clear calibration narrative (e.g., Wilks, 1938; Cox & Hinkley, 1979; Royall, 2017; Edwards, 2018), thereby providing both principledness and transparency. What remains largely open is $(iii)$: *computational feasibility*. For modern large models (foundation models, large language models, and multi-modal systems), retraining ensembles or running costly Bayesian pipelines is often prohibitive. This work aims to close this gap by deriving efficient credal predictions while retaining the likelihood-based semantics. We first fix notation and briefly recall the (relative) likelihood-based construction.

Let $L : \mathcal{H} \to [0, \infty)$ denote the (empirical) likelihood of a hypothesis on $\mathcal{D}_{\text{train}}$, and let

$$\gamma(h) \ := \ \frac{L(h)}{\sup_{h \in \mathcal{H}} L(h)} \in [0, 1].$$

Thus $\gamma(h)$ is the *relative likelihood* (likelihood ratio) of $h$ with respect to a maximum-likelihood solution: $\gamma(h) = 1$ for any MLE (if it exists) and decreases as the fit worsens; equivalently, $\log \gamma(h)$ is the log-likelihood gap, and $-2 \log \gamma(h)$ is the usual likelihood-ratio statistic.

For $\alpha \in (0, 1]$, define the *plausible model set* (*viz.* relative-likelihood $\alpha$-cut)

$$\mathcal{C}_\alpha \ := \ \{ h \in \mathcal{H} : \gamma(h) \geq \alpha \}.$$

Given $\boldsymbol{x} \in \mathcal{X}$, the predictive image of $\mathcal{C}_\alpha$ is $\mathcal{Q}_{\boldsymbol{x}, \alpha} \ := \ \{ p(\cdot \mid \boldsymbol{x}, h) : h \in \mathcal{C}_\alpha \} \ \subseteq \ \Delta_K$. A convenient class-wise summary of $\mathcal{Q}_{\boldsymbol{x}, \alpha}$ is given by the marginal extrema

$$\underline{p}_k(\boldsymbol{x}) \ := \ \inf_{h \in \mathcal{C}_\alpha} p_k(\boldsymbol{x}, h), \qquad \overline{p}_k(\boldsymbol{x}) \ := \ \sup_{h \in \mathcal{C}_\alpha} p_k(\boldsymbol{x}, h), \quad k = 1, \dots, K. \tag{1}$$

We then define the *box credal set* at $\boldsymbol{x}$ as

$$\square_{\boldsymbol{x}, \alpha} \ := \ \Big\{ p \in \Delta_K \ : \ \underline{p}_k(\boldsymbol{x}) \leq p_k \leq \overline{p}_k(\boldsymbol{x}) \ \forall k \Big\}. \tag{2}$$

By construction, $\mathcal{Q}_{\boldsymbol{x}, \alpha} \subseteq \square_{\boldsymbol{x}, \alpha}$; thus, the box is a tractable outer approximation that preserves all classwise extrema. We state a simple, yet illustrative monotonicity property:

**Proposition 2.1.** *If $0 < \alpha_2 \leq \alpha_1 \leq 1$, then $\mathcal{C}_{\alpha_1} \subseteq \mathcal{C}_{\alpha_2}$ and $\mathcal{Q}_{\boldsymbol{x}, \alpha_1} \subseteq \mathcal{Q}_{\boldsymbol{x}, \alpha_2}$. Thus, for all $k$,*

$$\underline{p}_k(\boldsymbol{x}; \alpha_1) \ \geq \ \underline{p}_k(\boldsymbol{x}; \alpha_2) \quad \text{and} \quad \overline{p}_k(\boldsymbol{x}; \alpha_1) \ \leq \ \overline{p}_k(\boldsymbol{x}; \alpha_2).$$

*If a maximum-likelihood estimator $h^{\text{ML}} \in \mathcal{H}$ exists, then $\mathcal{Q}_{\boldsymbol{x}, 1} = \{ p_k(\boldsymbol{x}, h^{\text{ML}}) \}$ and $[\underline{p}_k(\boldsymbol{x}; 1), \overline{p}_k(\boldsymbol{x}; 1)] = \{ p_k(\boldsymbol{x}, h^{\text{ML}}) \}$. As $\alpha \downarrow 0$, $\mathcal{Q}_{\boldsymbol{x}, \alpha} \to \{ h(\boldsymbol{x}) : L(h) > 0 \}$, and the intervals expand accordingly to the coordinatewise infima/suprema over that limit set.*

Proposition 2.1 shows that increasing the plausibility threshold $\alpha$ yields nested prediction sets and monotonically tighter classwise intervals. This monotonicity underpins the so-called *coverage–efficiency* trade-off used in our evaluation: larger $\alpha$ typically lowers coverage but improves efficiency (smaller sets), allowing $\alpha$ to be tuned to the desired operating point. In this vein, it is natural to evaluate set-valued predictions along two axes, *coverage* and *efficiency*.

**Coverage.** Given a set-valued predictor $H$, coverage is the probability that the ground-truth conditional distribution $p^\star(\cdot \mid \boldsymbol{x})$ is contained in the predicted set:

$$C(H) \ = \ \mathbb{E}\big[ \mathbf{1}\{ p^\star(\cdot \mid \boldsymbol{x}) \in H(\boldsymbol{x}) \} \big], \tag{3}$$

where the expectation is over the marginal of $\boldsymbol{x}$ on $\mathcal{X}$.

**Efficiency.** To reward informative (i.e., small) sets, we use the complement of the average interval width across classes (positively oriented: higher is better):

$$E(H) \;=\; 1 - \mathbb{E}\left[ \frac{1}{K} \sum_{k=1}^{K} \big( \overline{p}_k(\boldsymbol{x}) - \underline{p}_k(\boldsymbol{x}) \big) \right]. \tag{4}$$

Pragmatically, constructing relative-likelihood $\alpha$-cuts by training ensembles to hit prescribed likelihood ratios is principled, but computationally heavy, and hypotheses tend to cluster near the MLE unless $\alpha \approx 1$, which is a poor fit for modern large models. To overcome this limitation, we propose an efficient method for credal prediction that is grounded in the notion of relative likelihood and inspired by techniques for the calibration of probabilistic classifiers, which we will call *decalibration*.

## 3 EFFICIENT CREDAL PREDICTION THROUGH DECALIBRATION

We propose *decalibration* as a single-model route to plausibility: starting from the maximum-likelihood predictor $h^{\mathrm{ML}}$, we deliberately distort predicted probabilities, thereby moving from better predictions (high likelihood) to worse ones (lower likelihood). However, to keep predictions plausible, we make sure to remain within a prescribed relative-likelihood budget $\alpha \in (0,1]$. The probabilities reachable under this budget induce, for any query $\boldsymbol{x}$, classwise plausible intervals and hence a box credal set, without retraining or ensembling. Broadly speaking, we answer the following question: to what extent can we decrease or increase the predicted probabilities for a specific class before reaching a state where predictions become unplausible (have relative likelihood $< \alpha$)?

Thus, the idea is to keep the likelihood semantics but make the search cheap: rather than training many "plausible" models, we start from the MLE and deliberately push its probabilities toward less likely configurations, while enforcing a global relative-likelihood budget $\alpha$, as illustrated in Figure 1. This turns the classical likelihood-ratio view ("models within an LR ball are plausible") into a post-hoc exploration of the model's output space. As the budget is imposed on the training likelihood, any probability vector we produce is still supported by the data up to the chosen evidence level.

Operationally, we implement the exploration with simple, low-dimensional transforms of the MLE's logits, expressive enough to traverse a wide range of alternative class probabilities yet requiring neither retraining nor access to the backbone's gradients. Compared to ensembles that approximate the same plausibility region by re-optimizing parameters, decalibration is orders of magnitude faster and model-agnostic, i.e., it works on top of any pretrained classifier. It is particularly suited to inference-only or API-gated systems, foundation models, LLMs, and multimodal encoders, where parameters are frozen or proprietary and retraining, fine-tuning, or ensembling is impractical or disallowed. At the same time, the outputs inherit clear interpretation, "probabilities reachable without losing more than an $\alpha$-fraction of likelihood", and the resulting intervals are nested as $\alpha$ increases.

Among many possible post–hoc maps on probabilities, we instantiate decalibration with a simple yet expressive family that operates on *logits*: we add a global bias vector $\boldsymbol{c} \in \mathbb{R}^K$ to every example's logits (both train and test) and then apply softmax. This choice is model–agnostic (no retraining or gradients), preserves the learned representation, and induces a concave change in training log–likelihood, which in turn makes the $\alpha$–plausible set convex. Intuitively, $\boldsymbol{c}$ effects controlled *odds tilts* between classes; its softmax invariance under $\boldsymbol{c} \mapsto \boldsymbol{c} + t\mathbf{1}$ yields a natural identifiability hyperplane and keeps optimization well posed as we shall demonstrate now. Formally, consider a predictor that produces logits $z^{(n)} \in \mathbb{R}^K$ on the training set and logits $z(\boldsymbol{x}) \in \mathbb{R}^K$ for any test point $\boldsymbol{x}$. Let $\boldsymbol{c} = (c_1, \ldots, c_K)^\top \in \mathbb{R}^K$ and define, for each training point $n$ and each class $j$,

$$p_j^{(n)}(\boldsymbol{c}) \;=\; \frac{\exp\big(z_j^{(n)} + c_j\big)}{\sum_{k=1}^{K} \exp\big(z_k^{(n)} + c_k\big)}, \qquad p_j(\boldsymbol{x}; \boldsymbol{c}) \;=\; \frac{\exp\big(z_j(\boldsymbol{x}) + c_j\big)}{\sum_{k=1}^{K} \exp\big(z_k(\boldsymbol{x}) + c_k\big)}.$$

Set $\Delta\ell(\boldsymbol{c}) \;=\; \sum_{n=1}^{N} \Big( \log p_{y^{(n)}}^{(n)}(\boldsymbol{c}) - \log p_{y^{(n)}}^{(n)}(0) \Big)$ and $F(\alpha) \;=\; \big\{ \boldsymbol{c} \in \mathbb{R}^K : \Delta\ell(\boldsymbol{c}) \geq \log \alpha \big\}$. Further, note that $\Delta\ell(\boldsymbol{c} + t\mathbf{1}) = \Delta\ell(\boldsymbol{c})$ and $p_j(\boldsymbol{x}; \boldsymbol{c} + t\mathbf{1}) = p_j(\boldsymbol{x}; \boldsymbol{c})$ for all $t \in \mathbb{R}$.

Concretely, this decalibration procedure fixes the family and its feasibility region $F(\alpha)$; what follows establishes the key structural properties that make the method tractable: smooth concavity of $\Delta\ell$, convexity/compactness of the feasible set, and a convex–optimization characterization of the

upper credal bound (Proposition 3.1). We then specialize to the one–dimensional slice $\boldsymbol{c} = t\,\mathbf{e}_k$, yielding endpoint formulas and simple convex programs for the scalar case (Corollary 3.1).

**Proposition 3.1.** *Let $S := \{\boldsymbol{c} \in \mathbb{R}^K : \mathbf{1}^\top \boldsymbol{c} = 0\}$ be the identifiability hyperplane. Then:*

(a) *$\Delta\ell$ is $C^\infty$ and concave on $\mathbb{R}^K$, with $\nabla\Delta\ell(\boldsymbol{c}) = \sum_{n=1}^{N}\big(\mathbf{e}_{y^{(n)}} - p^{(n)}(\boldsymbol{c})\big)$ and*

$$\nabla^2\Delta\ell(\boldsymbol{c}) = -\sum_{n=1}^{N}\Big(\mathrm{Diag}\big(p^{(n)}(\boldsymbol{c})\big) - p^{(n)}(\boldsymbol{c})\,p^{(n)}(\boldsymbol{c})^\top\Big) \preceq 0.$$

*Moreover, $\Delta\ell$ is strictly concave on $S$ provided at least two classes appear in $\mathcal{D}$, namely, $N_j > 0$ for at least two $j$, where $N_j = \#\{n : y^{(n)} = j\}$. Consequently, $F(\alpha)$ is convex and translation-invariant along $\mathrm{span}\{\mathbf{1}\}$. The section $F_S(\alpha) := F(\alpha) \cap S$ is nonempty and compact whenever at least two classes appear.*

(b) *For each fixed $\boldsymbol{x}$ and $k$, the map $\boldsymbol{c} \mapsto \log p_k(\boldsymbol{x}; \boldsymbol{c}) = (z_k(\boldsymbol{x}) + c_k) - \log\sum_{l=1}^{K} e^{z_l(\boldsymbol{x})+c_l}$ is $C^\infty$ and concave on $\mathbb{R}^K$ with $\nabla\log p_k(\boldsymbol{x}; \boldsymbol{c}) = \mathbf{e}_k - p(\boldsymbol{x}; \boldsymbol{c})$ and*

$$\nabla^2\log p_k(\boldsymbol{x}; \boldsymbol{c}) = -\Big(\mathrm{Diag}\big(p(\boldsymbol{x}; \boldsymbol{c})\big) - p(\boldsymbol{x}; \boldsymbol{c})\,p(\boldsymbol{x}; \boldsymbol{c})^\top\Big) \preceq 0.$$

*In particular, $c_k \mapsto p_k(\boldsymbol{x}; \boldsymbol{c})$ is strictly increasing (holding $c_j$, $j \neq k$, fixed), and $c_j \mapsto p_k(\boldsymbol{x}; \boldsymbol{c})$ is strictly decreasing for $j \neq k$.*

(c) *The upper credal bound is the value of the convex optimization problem*

$$\bar{p}_k(\boldsymbol{x}) = \sup_{\boldsymbol{c}\in F(\alpha)} p_k(\boldsymbol{x}; \boldsymbol{c}) = \sup_{\boldsymbol{c}\in F(\alpha)} \exp\big(\log p_k(\boldsymbol{x}; \boldsymbol{c})\big) = \exp\Big(\sup_{\boldsymbol{c}\in F(\alpha)} \log p_k(\boldsymbol{x}; \boldsymbol{c})\Big),$$

*and the inner problem $\sup_{\boldsymbol{c}\in F(\alpha)} \log p_k(\boldsymbol{x}; \boldsymbol{c})$ is a concave maximization, i.e., a convex optimization. An optimizer always exists on $F_S(\alpha)$, and is unique modulo addition of constants along $\mathrm{span}\{\mathbf{1}\}$.*

(d) *The lower credal bound $\underline{p}_k(\boldsymbol{x}) = \inf_{\boldsymbol{c}\in F(\alpha)} p_k(\boldsymbol{x}; \boldsymbol{c})$ is, in general, not a convex optimization problem. Nevertheless, when $F_S(\alpha)$ is compact, a minimizer exists and is attained at an extreme point of the convex set $F_S(\alpha)$.*

We prove Proposition 3.1 in Appendix A. Proposition 3.1 (a) guarantees that the likelihood-based feasibility region is a well-posed convex set (compact on the hyperplane $S$), so optimization over it is stable. Moreover, (b) shows the test objective inherits the same curvature structure as the training likelihood. Together these yield (c): the *upper* credal bound is the value of a single convex program with a unique optimizer on $S$, while (d) clarifies that the *lower* bound is generally nonconvex and lives on the boundary/extreme points of $F_S(\alpha)$. Practically, the upper credal bounds can be computed reliably using convex solvers, while the lower bounds require exploration of the feasibility set's boundary. This task, however, is not trivial: the lower-bound optimization may feature multiple global extrema. Thoroughly exploring these extrema can become computationally expensive, undermining the main goal of our approach, which is to efficiently compute credal predictions.

To address this challenge we restrict to class-specific biases $\boldsymbol{c} = t\,\mathbf{e}_k$. This reduces the problem to a tractable one-dimensional slice, where the feasible set becomes a simple interval, and the class-$k$ probability varies monotonically with $t$. Consequently, exact lower and upper bounds can be obtained simply by evaluating the interval endpoints, as we formalize in the following corollary.

**Corollary 3.1.** *Now restrict to shifts of the form $\boldsymbol{c} = t\,\mathbf{e}_k$, $t \in \mathbb{R}$ and define*

$$\Delta\ell_k(t) := \Delta\ell(t\,\mathbf{e}_k), \qquad F_k(\alpha) := \{t \in \mathbb{R} : \Delta\ell_k(t) \geq \log\alpha\} = \{t : t\,\mathbf{e}_k \in F(\alpha)\}.$$

*Then the following hold:*

(a) *$\Delta\ell_k$ is $C^2$ and strictly concave on $\mathbb{R}$. Consequently $F_k(\alpha)$ is a nonempty interval; if $0 < N_k < N$, it is compact $[t_k^-, t_k^+]$, otherwise it is a closed (possibly half-infinite) interval.*

(b) *For every fixed $\boldsymbol{x}$, the map $t \mapsto p_k(\boldsymbol{x}; t\,\mathbf{e}_k)$ is strictly increasing on $\mathbb{R}$.*

(c) *With $t_k^- = \inf F_k(\alpha)$ and $t_k^+ = \sup F_k(\alpha)$,*

$$\underline{p}_k(\boldsymbol{x}) = p_k\big(\boldsymbol{x}; t_k^-\, \mathbf{e}_k\big), \qquad \overline{p}_k(\boldsymbol{x}) = p_k\big(\boldsymbol{x}; t_k^+\, \mathbf{e}_k\big).$$

(d) *The endpoints $t_k^-, t_k^+$ solve the convex programs*

$$\min_{t\in\mathbb{R}}\ t\ \ s.t.\ -\Delta\ell_k(t) \leq -\log\alpha, \qquad \min_{t\in\mathbb{R}}\ (-t)\ \ s.t.\ -\Delta\ell_k(t) \leq -\log\alpha.$$

We prove Corollary 3.1 in Appendix A. Algorithmically, the scalar case reduces computing $\big(\underline{p}_k(\boldsymbol{x}), \overline{p}_k(\boldsymbol{x})\big)$ to finding the two endpoints $t_k^-$ and $t_k^+$ of the feasible interval, e.g., by bisection on $\Delta\ell_k(t) = \log\alpha$; the bounds are then $p_k(\boldsymbol{x}; t_k^-\mathbf{e}_k)$ and $p_k(\boldsymbol{x}; t_k^+\mathbf{e}_k)$. Throughout the empirical evaluation, we focus on the one-dimensional setting, which admits convexity of the lower and upper probability bounds, resulting in the box credal set $\square_{\boldsymbol{x},\alpha}$. We defer details about the practical computation of the bounds to Appendix B.

## 4 EMPIRICAL RESULTS

In this section, we empirically evaluate our proposed method with the following research objectives in mind. First, we assess the quality of the uncertainty representation by standard metrics and show its strong performance compared to baselines in Section 4.1. Second, we evaluate the method on common downstream tasks and emphasize the competitive performance—while far more efficient— when compared to baselines in Section 4.2. Third, we highlight the distinctive advantage of our method that it can construct uncertainty representations for large architectures such as `TabPFN` or `CLIP`, where retraining is infeasible in Sections 4.3 and 4.4.

Thus, we present scenarios where our method (EffCre, see Appendix B for implementation details) newly enables the construction of uncertainty representations and, where possible, we compare it to the following suitable baselines, which represent the current state-of-the-art in credal prediction: Credal Wrapper (CreWra) (Wang et al., 2025a), Credal Ensembling (CreEns) (Nguyen et al., 2025), Credal Bayesian Neural Networks (CreBNN) (Caprio et al., 2024a), Credal Interval Net (CreNet) (Wang et al., 2025b), and Credal Relative Likelihood (CreRL) (Löhr et al., 2025). The code for all experiments is published in a Github repository[1] and the detailed experimental setup can be found in Appendix D.

### 4.1 COVERAGE VERSUS EFFICIENCY

We compare our method to the baselines in terms of coverage (3) and efficiency (4). Ideally, a credal predictor generates sets of a small size (high efficiency) that cover the ground-truth conditional distribution (high coverage). Moreover, because the relative importance of coverage and efficiency may vary across applications, methods that allow to trade-off one against the other, depending on the setting, are favored.

**Setup.** We train a multilayer perceptron (`MLP`) on the embeddings of CHAOSNLI and a `ResNet18` (He et al., 2016) on CIFAR-10 (Krizhevsky et al., 2009) (Schmarje et al., 2022). The models are trained with regular labels and evaluated against ground-truth distributions. Such ground-truth distributions are derived from multiple annotator labels, available through CIFAR-10H (Peterson et al., 2019) for the CIFAR-10 test set, while CHAOSNLI provides them directly.

**Results.** We present results for coverage versus the efficiency in Figure 2. The CIFAR-10 dataset shows that our method Pareto dominates CreRL in the high coverage region, while performing similarly in the medium coverage region. In addition, EffCre Pareto dominates the CreBNN, CreWra, CreNet baselines. For the CHAOSNLI dataset our method performs similarly to CreRL in the high coverage region and similar to CreEns in the low coverage region. Whereas the aforementioned baselines can only traverse the low coverage *or* high coverage regions, our method can traverse both regions, allowing a user to specify almost any coverage or efficiency value. Furthermore, our method dominates CreBNN with $\alpha = 0.95$, whereas it performs similarly to CreNet and CreWra, again with the caveat that these methods are restricted to the low coverage region. For results on an

---

[1] https://github.com/pwhofman/efficient-credal-prediction.

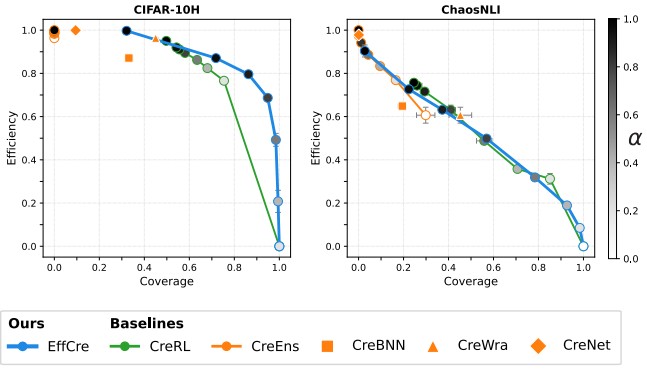

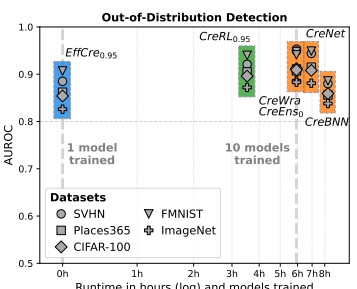

**Figure 2: Coverage versus Efficiency.** Comparison on CIFAR-10 and CHAOSNLI. The plot highlights the Pareto trade-off: higher coverage often requires lower efficiency. EffCre consistently advances the Pareto front over baselines.

**Figure 3: Out-of-Distribution Detection.** Performance (AUROC, based on epistemic uncertainty) as a function of required number of models and training time (in hours).

additional dataset, we refer to Appendix E.1. Lastly, note that, while in Section 2, we assume $\alpha > 0$, in the coverage and efficiency experiments we explicitly use $\alpha = 0$ as a verification that our method produces sufficiently diverse sets. Broadly speaking, if our method is not able to generate dense credal sets for $\alpha = 0$, we cannot expect it to reliably reach the edges of the plausible probability intervals.

## 4.2 OUT-OF-DISTRIBUTION DETECTION

Besides coverage and efficiency, out-of-distribution detection is a commonly used proxy to evaluate the quality of credal predictions. Modern machine learning systems should be able to detect whether the data they receive is in-distribution (ID) or out-of-distribution (OOD) as strong performance on this task is an indication of a good epistemic uncertainty representation. So far, many approaches have been unable to provide epistemic uncertainty representation due to prohibitive computational costs; our method directly addresses this. To demonstrate this, we analyze the trade-off between the training time and performance, in terms of AUROC, for our method and compare it to baselines.

**Setup.** We train a `ResNet18` on CIFAR-10, which serves as the ID data and introduce it to several other datasets that serve as the OOD data. Epistemic uncertainty is quantified based on a commonly-used measure from Abellán et al. (2006),

$$\text{EU}(\mathcal{Q}_{\boldsymbol{x}}) \coloneqq \overline{\text{S}}(\mathcal{Q}_{\boldsymbol{x}}) - \underline{\text{S}}(\mathcal{Q}_{\boldsymbol{x}}), \tag{5}$$

where $\overline{\text{S}}(\mathcal{Q}_{\boldsymbol{x}}) = \sup_{p(\cdot \mid \boldsymbol{x}) \in \mathcal{Q}_{\boldsymbol{x}}} \text{S}(p(\cdot \mid \boldsymbol{x}))$, and $\underline{\text{S}}(\mathcal{Q}_{\boldsymbol{x}})$ defined analogously, are the upper and lower Shannon entropy[2], respectively.

**Results.** We report the OOD detection results alongside training time in Figure 3 and provide additional results and hyperparameter ablations in Appendix F.1. Since any approach requires at least one trained model (e.g., an MLE predictor), our method EffCre comes with no extra training cost, as it can be applied post-hoc in a highly efficient manner. Although the baselines achieve slightly higher AUROC scores on the OOD task, they rely on ensembles of models, which demand a substantial number of members (e.g., 10 models) and therefore significantly increase training time. While CreWra, CreEns, CreNet, and CreBNN require full training of each ensemble member, CreRL is slightly more efficient due to its early-stopping criterion. However, our approach EffCre is substantially more efficient compared to all baselines, enabling the application even to large-scale models.

## 4.3 IN-CONTEXT LEARNING WITH TABPFN

To highlight the ability of our method to be applied in a *post-hoc* manner, requiring only logits, we apply it to a foundation model for tabular data.

---

[2]Shannon entropy: $\text{S}(p(\cdot \mid \boldsymbol{x})) = -\sum_{k=1}^{K} p_k(\boldsymbol{x}) \log p_k(\boldsymbol{x})$ with $0 \log 0 = 0$ by definition.

TabPFN (Hollmann et al., 2025) is a prior-fitted transformer, trained on a large number of synthetic datasets. It uses in-context learning, based on all training data and additional exemplary instances, to make predictions, while not requiring any gradient-based changing of its weights. Therefore, the baselines used in the experiments in Sections 4.1 and 4.2 cannot be applied as they require training (an ensemble), which, besides being challenging due to computational cost, also requires the *original* training data, which we do not have access to.

**Setup Coverage Versus Efficiency.** To illustrate the proper uncertainty representation generated by using our method on top of TabPFN, we compute the coverage and efficiency of the predicted credal sets by applying it to all multi-class datasets[3] from the TABARENA benchmark (Erickson et al., 2025). Since these datasets do not provide ground-truth conditional distributions, we propose a simple way to create *semi-synthetic ground-truth distributions* to allow evaluation by coverage and efficiency. Details about this experiment and the process of creating such distributions can be found in Appendix E.3 and Appendix D.5, respectively.

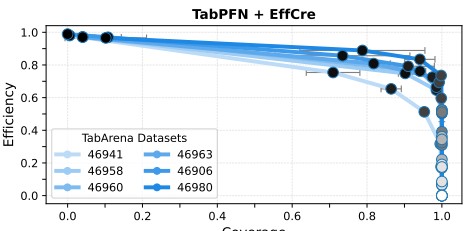

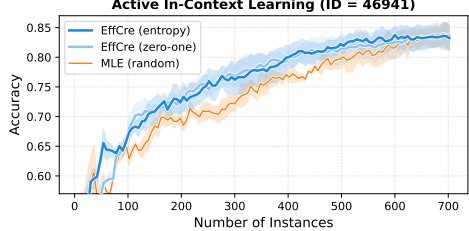

**Figure 4: EffCre used with TabPFN. Top:** Coverage versus efficiency performance all multi-class TABARENA datasets. **Bottom:** Active In-Context Learning performance versus the random baseline.

**Results Coverage Versus Efficiency.** We show the coverage and efficiency results in Figure 4, confirming that uncertainty representations obtained by applying our method to TabPFN provide small sets that often include the ground-truth distribution.

In addition, we perform active in-context learning, which has become an important task in the context of foundation models since labeling represents the limiting factor to leveraging pre-trained models effectively. Ideally, a model—equipped with a reliable (epistemic) uncertainty representation—is able to sample informative instances, which, when used for in-context learning, improve the performance more than a random sample of instances would.

**Setup Active Learning.** Specifically, we quantify epistemic uncertainty using (5) and additionally use a measure based on zero-one-loss, which has been show to perform well on similar tasks (Hofman et al., 2024). Concretely,

$$\mathrm{EU}(\mathcal{Q}_{\boldsymbol{x}}) = \max_{p(\cdot|\boldsymbol{x}),\, p'(\cdot|\boldsymbol{x}) \in \mathcal{Q}_{\boldsymbol{x}}} \max_k p_k(\boldsymbol{x}) - p_{\arg\max_k p'_k(\boldsymbol{x})}(\boldsymbol{x}). \tag{6}$$

We perform this task for a number of TABARENA datasets using $\alpha = 0.8$. For more details regarding the experimental setup and additional results, we refer to Appendix E.3.

**Results Active Learning.** We present the results in Figure 4. This highlights the ability of our method to represent its epistemic uncertainty well, in order to sample the most informative instances accordingly. An ablation on $\alpha$ in the active in-context learning setting can be found in Appendix F.2.

## 4.4 ZERO-SHOT CLASSIFICATION WITH CLIP-BASED MODELS

We demonstrate the flexibility of our method by creating credal sets for vision–language models (VLMs), including CLIP (Radford et al., 2021), SigLIP (Zhai et al., 2023), SigLIP-2 (Tschannen et al., 2025), and BiomedCLIP (Zhang et al., 2024).

**Setup Coverage Versus Efficiency.** To demonstrate the proper uncertainty representation generated by using our method on top of CLIP-based models—something that is computationally prohibitive for the baselines—we compute the coverage and efficiency of the predicted credal sets by applying it to CIFAR-10, using CIFAR-10H as ground-truth distributions. Therefore, we turn the models into zero-shot classifiers by reformulating the label set into natural-language templates and comparing

---

[3]The datasets included in TabArena v0.1. This collection may be subject to change.

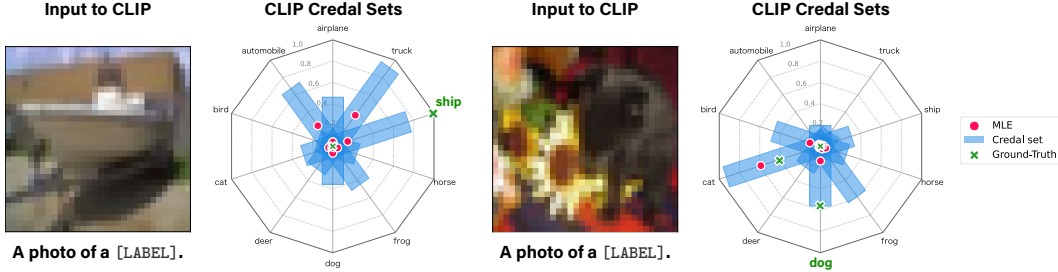

**Figure 5: Credal Prediction with `CLIP`.** Examples from CIFAR-10H with high epistemic uncertainty (**left**) and high aleatoric uncertainty (**right**) as predicted by applying our method on `CLIP`.

the resulting text embeddings with image embeddings (see Appendix D.6 for details and performance results).

**Results Coverage Versus Efficiency.** Figure 6 shows performance on the coverage-efficiency trade-off, with our method performing well, being able to reach regions with high coverage and high efficiency for `CLIP`, `SigLIP`, and `SigLIP-2`.

To further highlight the usefulness of our method, we compare the predicted credal sets to human uncertainty patterns. To visualize credal sets beyond three classes, we propose *credal spider plots* where each axis corresponds to a class and intervals mark upper and lower probabilities (see Appendix C for a detailed guide). Profiles such as MLE predictions or ground-truth distributions can be overlaid for direct comparison.

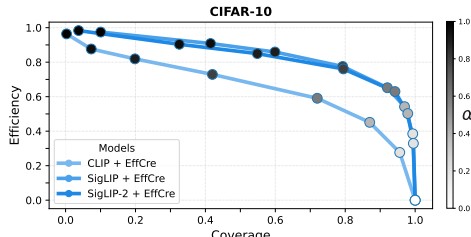

**Setup Qualitative Evaluation.** For the visual evaluation, we apply our method on top of `CLIP`-based models and predict credal sets for CIFAR-10 while using CIFAR-10H to reference ground-truth distributions. We sort and present instances from CIFAR-10-H's test split based on the aleatoric and epistemic uncertainties represented by the predicted credal sets of the `CLIP`-based models.

**Figure 6: EffCre used with `CLIP`-based models.** We demonstrate the performance of EffCre by applying it to `CLIP`, `SigLIP`, and `SigLIP-2` without retraining—something that can't be done by the baselines.

**Results Qualitative Evaluation.** Figure 5 presents two instances from CIFAR-10-H's test split: the left image is misclassified by the MLE due to the unusual context of the ship being out of water in a dock. Our method reflects the resulting epistemic uncertainty with plausible intervals across all classes and higher probability intervals for class ship, truck, and automobile. The right image shows an animal in an ambiguous pose: the ground-truth distribution splits mass between dog and cat, representing aleatoric uncertainty. Our method represents this uncertainty well, covering the true distribution even though the MLE misclassifies the image as cat. Additional multilingual examples and further results are shown in Appendix E.4.

## 5    RELATED WORK

Credal sets originate in imprecise-probability literature (Levi, 1978; Walley, 1991). In machine learning, credal sets offer an appealing way to represent epistemic uncertainty, motivating work on (credal) uncertainty quantification (Hüllermeier et al., 2022; Sale et al., 2023; Hofman et al., 2024; Chau et al., 2025a), calibration (Jürgens et al., 2025; Chau et al., 2025b), self-supervision (Lienen & Hüllermeier, 2021), and learning theory (Caprio et al., 2024b). A consensus on construction has not emerged, current practice spans a variety of designs that trade theoretical guarantees against practicality in different ways. Some methods use conformal prediction to obtain credal sets with finite-sample validity guarantees (Javanmardi et al., 2024). Others form sets by aggregating multiple predictors, whether standard deep ensembles (Wang et al., 2025a; Nguyen et al., 2025), interval-

head networks trained with tailored losses (Wang et al., 2025b), or Bayesian ensembles built from posterior samples (Caprio et al., 2024a). Recently, Löhr et al. (2025) adopted a relative-likelihood criterion (Birnbaum, 1962; Antonucci et al., 2012; Senge et al., 2014) to form credal sets in machine learning settings.

Ensemble training underpins much of the prior work, but for large model architectures, retraining—even once—is rarely feasible. This, in turn, has led to lightweight alternatives for representing uncertainty. One line of work focuses on single forward pass methods to estimate uncertainty (van Amersfoort et al., 2020; Mukhoti et al., 2023), e.g., via distance-based features (Liu et al., 2020) or evidential Dirichlet heads (Sensoy et al., 2018; Amini et al., 2020), though recently evidential variants were criticized (Bengs et al., 2023; Jürgens et al., 2024). In contrast, approximate Bayesian inference techniques such as Laplace approximations (Daxberger et al., 2021; Weber et al., 2025) and variational last-layer or sub-ensemble approaches (Valdenegro-Toro, 2019; Kristiadi et al., 2020; Harrison et al., 2024) reduce cost by training only limited parts of the network. Similarly, others compress ensembles by sharing parameters (Durasov et al., 2021; Laurent et al., 2023) or by distilling ensemble knowledge into a single model (Malinin et al., 2020; Penso et al., 2022). Another line of work focuses on large models such as diffusion or language models and explores low-rank adaptation to efficiently build ensembles for uncertainty quantification (Berry et al., 2024; Yang et al., 2024; Wang et al., 2024). Yet, computationally efficient methods for credal prediction remain absent from the literature.

## 6 DISCUSSION

We presented a post-hoc, model-agnostic method for credal prediction that captures epistemic uncertainty as class-wise *plausible probability intervals* derived from relative likelihood. The key idea, *decalibration*, perturbs a trained model's logits under a global likelihood-ratio budget, thereby exploring less-likely yet still plausible predictions without retraining. We formally analyze decalibration and show that the logit-shift feasibility set is convex (compact on an identifiability hyperplane). In the one-logit (class-specific) case, each interval endpoint is obtained by solving a small convex program, readily handled by off-the-shelf optimizers. Empirically, our method matches or surpasses baselines on coverage–efficiency and is competitive for OOD detection, while cutting computation by orders of magnitude. Because it is post-hoc and needs only logits, we apply it to large pretrained models—including `TabPFN` and `CLIP`—for which ensemble retraining is impractical.

**Limitations and Future Work.** We primarily deploy the one-logit (class-specific) variant of our logit-shift family. The fully coupled case remains open; upper bounds still reduce to a convex program, whereas lower bounds are non-convex. Developing reliable relaxations, certificates, or approximation schemes, and clarifying their statistical trade-offs, is a promising direction. Open-vocabulary, multimodal models such as `CLIP` raise additional challenges. Because the label set is chosen at inference time, uncertainty should reflect not only prediction but also label selection and prompt choice. Designing credal formalisms and evaluation protocols for this setting is an important avenue for future work.

**Reproducibility Statement.** We are committed to ensuring the reproducibility of our results. To this end, we provide our **code** in the following Github repository https://github.com/pwhofman/efficient-credal-prediction. The **theoretical results** in Section 3 are accompanied by proofs in Appendix A and, where necessary, the assumptions have been discussed. The full **experimental setup**, used to produce the results presented in Section 4 and Appendices E and F, is provided in Appendix D. In particular, we discuss details about **datasets**, including the transformation performed on the input to models and the creation of (semi-synthetic) ground truth distributions in Appendices D.2 and D.5. The **models** we use, and our implementation of them, are discussed in detail in Appendix D.1. We elaborate on the implementation of all **baselines** in Appendix D.3 and details regarding the practical implementation of **our method**, that are not discussed in the main paper, are provided in Appendix B.

ACKNOWLEDGMENTS

Timo Löhr and Maximilian Muschalik gratefully acknowledge funding by the Deutsche Forschungs-gemeinschaft (DFG, German Research Foundation): TRR 318/3 2026 – 438445824. Yusuf Sale is supported by the DAAD program Konrad Zuse Schools of Excellence in Artificial Intelligence, sponsored by the Federal Ministry of Education and Research.

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

## ORGANIZATION OF THE APPENDIX

We structure the appendix as follows: Appendix A provides a proof for Proposition 2.1 and Proposition 3.1, followed by a detailed description of our implementation in Appendix B. Appendix C describes the newly introduced *credal spider plots* in detail, before we give details about the different setups of our experiments in Appendix D. We finish with additional results in Appendix E and two ablation studies in Appendix F.

## A PROOFS

*Proof of Proposition 2.1.* Write $\gamma(h) = L(h)/\sup_{g \in \mathcal{H}} L(g) \in [0, 1]$, $\mathcal{C}_\alpha = \{h \in \mathcal{H} : \gamma(h) \geq \alpha\}$, and $\mathcal{Q}_{\boldsymbol{x},\alpha} = \{p(\cdot \mid \boldsymbol{x}, h) : h \in \mathcal{C}_\alpha\}$. If $0 < \alpha_2 \leq \alpha_1 \leq 1$ and $h \in \mathcal{C}_{\alpha_1}$, then $\gamma(h) \geq \alpha_1 \geq \alpha_2$, hence $h \in \mathcal{C}_{\alpha_2}$. Thus $\mathcal{C}_{\alpha_1} \subseteq \mathcal{C}_{\alpha_2}$ and, by applying the prediction map, $\mathcal{Q}_{\boldsymbol{x},\alpha_1} \subseteq \mathcal{Q}_{\boldsymbol{x},\alpha_2}$. Consequently, for each class $k$, $\underline{p}_k(\boldsymbol{x}; \alpha_1) = \inf_{h \in \mathcal{C}_{\alpha_1}} p_k(\boldsymbol{x}, h) \geq \inf_{h \in \mathcal{C}_{\alpha_2}} p_k(\boldsymbol{x}, h) = \underline{p}_k(\boldsymbol{x}; \alpha_2)$, and similarly $\overline{p}_k(\boldsymbol{x}; \alpha_1) = \sup_{h \in \mathcal{C}_{\alpha_1}} p_k(\boldsymbol{x}, h) \leq \sup_{h \in \mathcal{C}_{\alpha_2}} p_k(\boldsymbol{x}, h) = \overline{p}_k(\boldsymbol{x}; \alpha_2)$.

If an MLE $h^{\mathrm{ML}}$ exist, then $\gamma(h^{\mathrm{ML}}) = 1$ and $\mathcal{C}_1$ is the set of MLEs. In particular, if the (predictive) MLE is unique at $\boldsymbol{x}$ (e.g., the MLE is unique, or all MLEs agree at $\boldsymbol{x}$), then

$$\mathcal{Q}_{\boldsymbol{x},1} = \{p(\cdot \mid \boldsymbol{x}, h^{\mathrm{ML}})\} \quad \text{and} \quad [\underline{p}_k(\boldsymbol{x}; 1), \overline{p}_k(\boldsymbol{x}; 1)] = \{p_k(\boldsymbol{x}, h^{\mathrm{ML}})\}.$$

As $\alpha \downarrow 0$, the sets $\mathcal{C}_\alpha$ increase to $\{h \in \mathcal{H} : L(h) > 0\}$, hence $\mathcal{Q}_{\boldsymbol{x},\alpha} \uparrow \{p(\cdot \mid \boldsymbol{x}, h) : L(h) > 0\}$. For an increasing family of sets, coordinate-wise infima over $\mathcal{Q}_{\boldsymbol{x},\alpha}$ decrease to the infimum over the union, and suprema increase to the supremum. Therefore,

$$\underline{p}_k(\boldsymbol{x}; \alpha) \downarrow \inf_{\{h:L(h)>0\}} p_k(\boldsymbol{x}, h), \qquad \overline{p}_k(\boldsymbol{x}; \alpha) \uparrow \sup_{\{h:L(h)>0\}} p_k(\boldsymbol{x}, h),$$

as claimed. This completes the proof. $\qquad\square$

*Proof of Proposition 3.1.* (a) For each $n$, define $\phi_n(\boldsymbol{c}) := \log p_{y^{(n)}}^{(n)}(\boldsymbol{c})$. Then

$$\phi_n(\boldsymbol{c}) = \langle \mathbf{e}_{y^{(n)}}, \boldsymbol{c} \rangle - \log \sum_{l=1}^K \exp(z_l^{(n)} + c_l) + \mathrm{const}(z^{(n)}),$$

hence $\phi_n$ is $C^\infty$. Direct differentiation yields

$$\nabla \phi_n(\boldsymbol{c}) = \mathbf{e}_{y^{(n)}} - p^{(n)}(\boldsymbol{c}), \qquad \nabla^2 \phi_n(\boldsymbol{c}) = -\Big(\mathrm{Diag}\big(p^{(n)}(\boldsymbol{c})\big) - p^{(n)}(\boldsymbol{c})\, p^{(n)}(\boldsymbol{c})^\top\Big) \preceq 0,$$

so each $\phi_n$ is concave, and thus $\Delta\ell(\boldsymbol{c}) = \sum_n \phi_n(\boldsymbol{c}) - \sum_n \phi_n(0)$ is $C^\infty$ and concave. The Hessian matrices in the sum are positive semi-definite with nullspace containing $\mathrm{span}\{\mathbf{1}\}$, since $\big(\mathrm{Diag}(p) - pp^\top\big)\mathbf{1} = 0$; therefore $\nabla^2 \Delta\ell(\boldsymbol{c}) \prec 0$ on $S$ provided at least two classes appear. Concavity implies that every level-set $\{\boldsymbol{c} : \Delta\ell(\boldsymbol{c}) \geq \tau\}$ is convex. Non-emptiness follows from $\Delta\ell(0) = 0 \geq \log\alpha$.

To see compactness of $F_S(\alpha)$ when at least two classes appear, fix $\boldsymbol{d} \in S \setminus \{0\}$ and consider $\boldsymbol{c} = t\boldsymbol{d}$ with $t \to \infty$. Then

$$\Delta\ell(t\boldsymbol{d}) = \sum_{n=1}^N \Big(\langle \mathbf{e}_{y^{(n)}}, t\boldsymbol{d} \rangle - \log \sum_{l=1}^K e^{z_l^{(n)} + td_l}\Big) + \mathrm{const} = t\sum_{j=1}^K N_j d_j - \sum_{n=1}^N \log \sum_{l=1}^K e^{z_l^{(n)} + td_l} + \mathrm{const}.$$

As $t \to \infty$, $\log \sum_l e^{z_l^{(n)} + td_l} = t\max_l d_l + O(1)$, hence

$$\Delta\ell(t\boldsymbol{d}) = t\Big(\sum_{j=1}^K N_j d_j - N\max_l d_l\Big) + O(1).$$

Because $\boldsymbol{d} \in S$ and at least two $d_l$ differ, we have $\sum_j N_j d_j < N\max_l d_l$. Thus $\Delta\ell(t\boldsymbol{d}) \to -\infty$ along every ray in $S$, proving coercivity on $S$, and hence compactness of $F_S(\alpha)$.

(b) Follows by differentiating

$$\log p_k(\boldsymbol{x}; \boldsymbol{c}) = (z_k(\boldsymbol{x}) + c_k) - \log \sum_{l=1}^K e^{z_l(\boldsymbol{x}) + c_l}.$$

The gradient and Hessian are as stated, and negative semi-definiteness of the Hessian shows concavity. The coordinate-wise monotonicity is immediate from $\partial \log p_k / \partial c_i = \delta_{ik} - p_i(\boldsymbol{x}; \boldsymbol{c})$ together with $p_i(\boldsymbol{x}; \boldsymbol{c}) \in (0, 1)$.

(c) Since $\log p_k(\boldsymbol{x}; \cdot)$ is concave and $F(\alpha)$ is convex, $\sup_{\boldsymbol{c} \in F(\alpha)} \log p_k(\boldsymbol{x}; \boldsymbol{c})$ is a concave maximization, i.e., a convex optimization problem. Existence of an optimizer on $F_S(\alpha)$ follows from compactness; invariance along $\mathrm{span}\{\mathbf{1}\}$ yields uniqueness only modulo translations by $\mathbf{1}$. The equality between $\sup p_k$ and $\exp(\sup \log p_k)$ follows from strict monotonicity of the exponential.

(d) Because $\log p_k(\boldsymbol{x}; \cdot)$ is concave, minimizing $p_k$ is equivalent to minimizing a concave function, which is not a convex optimization problem in general. Nevertheless, on the compact convex set $F_S(\alpha)$, a minimizer exists and is attained at an extreme point by standard results on concave functions over convex compact sets.

This completes the proof. □

*Proof of Corollary 3.1.* Fix $k \in \{1, \dots, K\}$ and restrict the multivariate objects of Proposition 3.1 to the affine line $\boldsymbol{c} = t\,\mathbf{e}_k$, $t \in \mathbb{R}$.

(a) Since $\Delta\ell(\cdot)$ is concave on $\mathbb{R}^K$ by Proposition 3.1(a), its restriction $t \mapsto \Delta\ell_k(t)$ is concave on $\mathbb{R}$; non-emptiness follows from $\Delta\ell_k(0) = 0 \geq \log \alpha$. Moreover, if $0 < N_k < N$, then $\Delta\ell_k(t) \to -\infty$ as $t \to +\infty$ (terms with $y^{(n)} \neq k$ decay like $-t$) and as $t \to -\infty$ (terms with $y^{(n)} = k$ decay like $t$), so $F_k(\alpha)$ is a compact interval $[t_k^-, t_k^+]$. If $N_k \in \{0, N\}$, the same tail check shows $F_k(\alpha)$ is a closed (possibly half-infinite) interval.

(b) From Proposition 3.1(b), $\nabla \log p_k(\boldsymbol{x}; \boldsymbol{c}) = \mathbf{e}_k - p(\boldsymbol{x}; \boldsymbol{c})$. Along the line $\boldsymbol{c} = t\,\mathbf{e}_k$,

$$\frac{d}{dt} \log p_k\big(\boldsymbol{x}; t\big) = \big(\mathbf{e}_k - p(\boldsymbol{x}; t\,\mathbf{e}_k)\big)^\top \mathbf{e}_k = 1 - p_k\big(\boldsymbol{x}; t\big),$$

hence $\frac{d}{dt} p_k(\boldsymbol{x}; t) = p_k(\boldsymbol{x}; t)\big(1 - p_k(\boldsymbol{x}; t)\big) > 0$. Thus $t \mapsto p_k(\boldsymbol{x}; t)$ is strictly increasing.

(c) Since $F_k(\alpha)$ is an interval and $t \mapsto p_k(\boldsymbol{x}; t)$ is strictly increasing, the infimum/supremum over $F_k(\alpha)$ are attained at the endpoints:

$$\underline{p}_k(\boldsymbol{x}) = p_k\big(\boldsymbol{x}; t_k^-\big), \qquad \overline{p}_k(\boldsymbol{x}) = p_k\big(\boldsymbol{x}; t_k^+\big).$$

(d) The feasible set can be written as $\{t \in \mathbb{R} : -\Delta\ell_k(t) \leq -\log \alpha\}$, where $-\Delta\ell_k$ is convex. Minimizing $t$ (resp. $-t$) over this convex set is a convex program whose optimizer is exactly the left (resp. right) endpoint $t_k^-$ (resp. $t_k^+$). In the half-infinite cases $N_k \in \{0, N\}$ the same conclusions hold with the appropriate limits $t_k^- = -\infty, t_k^+ = +\infty$ (so $p_k(\boldsymbol{x}; t_k^-) = 0$ or $p_k(\boldsymbol{x}; t_k^+) = 1$).

This completes the proof. □

## B    IMPLEMENTATION DETAILS

In this section, we provide a detailed description of how our method EffCre is practically implemented, including the optimization procedure, the evaluation of probability intervals, and the steps taken to ensure computational efficiency.

As discussed in Section 3, given the logits of the MLE, our method essentially solves two convex optimization problems per class to determine the boundaries—namely, the lower and upper probabilities—of a plausible interval according to the relative likelihood constraint. Specifically, for each class logit of the MLE, we add a value to the logit to perturb the resulting probability, thereby deriving a bound for the plausible probability interval as a result of the optimization. In practice, we use the `minimize` function from SciPy (Virtanen et al., 2020) to optimize this value, with the relative likelihood threshold as a constraint, an initial solution of 0, and bounds set to $(-10000, 10000)$. Roughly speaking, each optimization produces a constant that is added to a single class logit, giving the lower (or upper) bound of the plausible probability interval for that class, which is used to construct the box credal set $\Box_{\boldsymbol{x},\alpha}$. As a result, applying our method to a dataset requires solving $2K$ convex optimization problems for each value of $\alpha$.

The constants obtained by our method, EffCre, can then be used to evaluate our method on test data instances, thus, constructing probability intervals, and thereby credal sets. Each interval bound is directly associated with a specific relative likelihood, which served as the constraint during the optimization. For models we trained ourselves, we use the training dataset to evaluate the log-likelihood and compare it with that of the maximum likelihood estimator (MLE) predictor to compute the relative likelihood, as described in Section 2. When the *original* training data is unavailable, we instead use a subset of the target dataset to compute the relative likelihood budget. For example, in the case of CLIP, we do not have access to the *original* training data, which spans many benchmark dataset in addition to a large sample of images from the internet. Since we want to make credal predictions for CIFAR-10, we use the respective train split of CIFAR-10 to compute the (relative) log-likelihoods in order to solve the optimization problem described above. Credal predictions are then made using the respective test split of the dataset.

In general, our setup allows for straightforward computation of alpha-cuts once results for multiple alpha values have been obtained, a task made feasible by the efficiency of our method. The implementation is simple and intuitive; for further clarity, we refer to the function `classwise_adding_optim_logit` in the code https://github.com/pwhofman/efficient-credal-prediction.

## C    GUIDE ON INTERPRETING CREDAL SPIDER PLOTS

So far, the quantitative evaluation of credal sets has mainly been restricted to the three class setting, due to the inability to visualize credal sets in a $(K-1)$-simplex for $K > 3$. As many machine learning problems involve more then three classes and as a visual representation of the output of models can give useful insight, it is important to be able to have such a visual representation. To enable this, we propose *credal spider plots*—these plots offer an intuitive way to evaluate the interval-based credal sets. Given an instance that we want to evaluate, we plot the different classes as variables in the spider diagram and generate bars, starting at the lower probability and ending at the upper probability, for each class, which represent the (plausible) probability intervals. The ground-truth distributions is then plotted as multiple dots (depending on the number of classes with non-zero probability mass) on the radii corresponding to the given probability mass of a class. As our method relies on the maximum likelihood estimate (MLE), we additionally plot the MLE in a similar way.

In Section 4.4, we sort instances in descending order by the aleatoric and epistemic uncertainty associated with the predicted credal set using measures by Abellán et al. (2006) that have been proposed on the basis of a number of suitable axioms. Specifically,

$$\underbrace{\overline{S}(\mathcal{Q}_{\boldsymbol{x}})}_{\mathrm{TU}(\mathcal{Q}_{\boldsymbol{x}}))} = \underbrace{\underline{S}(\mathcal{Q}_{\boldsymbol{x}})}_{\mathrm{AU}(\mathcal{Q}_{\boldsymbol{x}}))} + \underbrace{(\overline{S}(\mathcal{Q}_{\boldsymbol{x}}) - \underline{S}(\mathcal{Q}_{\boldsymbol{x}}))}_{\mathrm{EU}(\mathcal{Q}_{\boldsymbol{x}}))} \tag{7}$$

Therefore, maximum aleatoric uncertainty (lower entropy) will manifest itself in the credal spider plot for $K$ classes as having intervals that include $1/K$ for all $K$ classes. The maximum epistemic uncertainty (difference upper and lower entropy) is obtained by having similar plausible intervals as for aleatoric uncertainty, but additionally, the plausible interval for (at least) one class should admit a probability of 1. Besides, this instance-wise coverage and efficiency can also easily be observed from the credal spider plot by evaluating whether the ground-truth point fall into the plausible intervals and by considering the average length of the aforementioned intervals, respectively.

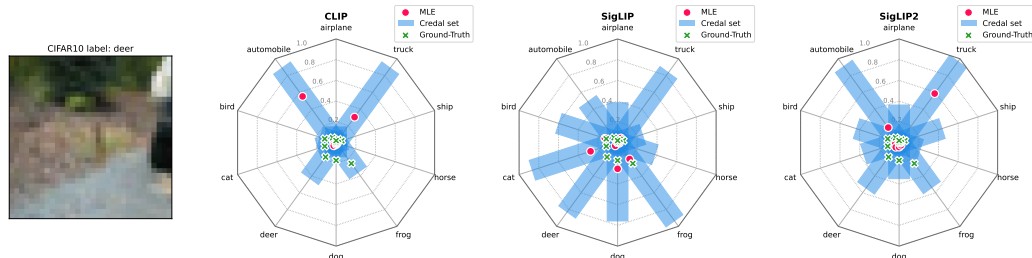

**Figure 7: Example credal spider plots for `CLIP`, `SigLIP`, and `SigLIP-2`.** used for illustration purposes. The credal spider plots includes the maximum likelihood estimate, the plausible intervals, and the ground-truth distribution.

# D EXPERIMENTAL SETUP

## D.1 MODELS

**Multilayer Perceptron** We train multilayer perceptron on the ChaosNLI dataset. The model consists of four linear layers with dimensions $[768 - 256 - 64 - 16 - 3]$, employing ReLU activations for all hidden layers, while the output layer uses a Softmax function to produce probability distributions from the logits. For training, we adopt hyperparameters similar to those identified as optimal in Javanmardi et al. (2024) (see Table 1).

**ResNet18** For experiments on CIFAR-10, we employ the ResNet-18 implementation and training configuration from https://github.com/kuangliu/pytorch-cifar. This variant is tailored to CIFAR-10 and is trained entirely from scratch, without ImageNet pretraining.

**TabPFN** TabPFN (Tabular Prior-data Fitted Network) (Hollmann et al., 2022) is a transformer-based foundation model developed for supervised classification and regression tasks on small to medium-sized tabular datasets, typically up to $10,000$ samples and $500$ features. Pre-trained on approximately 130 million synthetic datasets generated via structural causal models, TabPFN learns to approximate Bayesian inference through a single forward pass, eliminating the need for task-specific tuning. It adeptly handles numerical and categorical features, missing values, and outliers. We use TabPFN for all experiments with tabular data as presented in Section 4.3. The model is publicly accessible under a *custom license based on Apache 2.0*, which includes an enhanced attribution requirement.

**CLIP** CLIP (Contrastive Language–Image Pretraining) (Radford et al., 2021) is a multimodal neural network that learns visual concepts from natural language supervision. Trained on $400$ million image-text pairs sourced from the internet, CLIP can understand images in the context of natural language prompts, enabling zero-shot classification across various tasks without task-specific tuning. It employs a vision transformer architecture to process images and a causal language model to process text, aligning both modalities in a shared embedding space. This design allows CLIP to generalize to a wide range of visual tasks by interpreting textual descriptions directly. We employ CLIP to assess our method's performance in zero-shot classification tasks, demonstrating its applicability to large-scale models without the need for task-specific training. The model is publicly available under the MIT License, permitting both academic and commercial use.

**SigLIP** SigLIP (Sigmoid Loss for Language-Image Pretraining) (Zhai et al., 2023) is a multimodal vision-language model that enhances the CLIP framework by employing a pairwise sigmoid loss function instead of the traditional softmax loss. This modification allows for more efficient scaling to larger batch sizes while maintaining or improving performance at smaller batch sizes. SigLIP utilizes separate image and text encoders to generate representations for both modalities, aligning them in a shared embedding space. The model has demonstrated superior performance in zero-shot image classification tasks compared to CLIP, achieving an ImageNet zero-shot accuracy of $84.5\%$ with a batch size of $32,000$. Same as for CLIP, we demonstrate with SigLIP the ability to construct credal sets based on large-scale models. SigLIP is publicly available under the Apache 2.0 license, facilitating research and application in various domains.

**SigLIP-2** SigLIP-2 (Sigmoid Loss for Language-Image Pretraining 2) (Tschannen et al., 2025) is a multilingual vision-language encoder. Building upon the original SigLIP, SigLIP-2 integrates advanced pretraining techniques—including captioning-based pretraining, self-supervised losses (self-distillation and masked prediction), and online data curation—to enhance semantic understanding, localization, and dense feature extraction. The model demonstrates improved performance in zero-shot classification, image-text retrieval, and transfer tasks, particularly when extracting visual representations for Vision-Language Models. Notably, SigLIP-2 introduces a dynamic resolution variant, NaFlex, which supports multiple resolutions and preserves the native aspect ratio, making it suitable for applications sensitive to image dimensions. We use SigLIP-2 in a similar fashion as SigLIP and CLIP for large-scale experiments. The model is publicly available under the Apache 2.0 license, facilitating research and application across various domains.

**BiomedCLIP**  `BiomedCLIP` (Zhang et al., 2024) is a multimodal biomedical foundation model, pretrained on the PMC-15M dataset—a collection of 15 million figure-caption pairs extracted from over 4.4 million scientific articles in PubMed Central. Utilizing PubMedBERT as the text encoder and Vision Transformer as the image encoder, `BiomedCLIP` is tailored for biomedical vision-language processing through domain-specific adaptations. It has demonstrated state-of-the-art performance across various biomedical tasks, including cross-modal retrieval, image classification, and visual question answering, outperforming previous models such as BioViL in radiology-specific tasks like RSNA pneumonia detection. The model is publicly available under the Apache 2.0 license, facilitating research and application in the biomedical domain.

**Hyperparameters**  For certain experiments in our empirical evaluation, we use pre-trained (foundation) models, which do not require training and thus do not need hyperparameter specifications. In contrast, for the coverage-efficiency experiments on CIFAR-10, ChaosNLI, and QualityMRI, we train models from scratch using a dataset-specific set of hyperparameters, summarized in Table 1. Multiple configurations were evaluated, and the best-performing setup was selected individually for each dataset. To ensure comparability, all methods—our approach as well as the baselines—share the same hyperparameter settings within a given dataset. The only exception is CreBNN, which, when trained with the Adam optimizer (Kingma & Ba, 2015), requires a KL-divergence penalty of $1e - 7$ and weight decay set to zero. When instead using SGD combined with a cosine annealing learning rate schedule (Loshchilov & Hutter, 2017), CreBNN additionally needs a momentum of 0.9 to achieve stable training.

**Table 1:** Hyperparameters used for each dataset.

| Hyperparameter | ChaosNLI | CIFAR-10 | QualityMRI |
|---|---|---|---|
| Model | FCNet | ResNet18 | ResNet18 |
| Epochs | 300 | 200 | 200 |
| Learning rate | 0.01 | 0.1 | 0.01 |
| Weight decay | 0.0 | 0.0005 | 0.0005 |
| Optimizer | Adam | SGD | SGD |
| Ensemble members | 20 | 20 | 20 |
| LR scheduler | - | CosineAnnealing | CosineAnnealing |

## D.2 DATASETS

**ChaosNLI**  ChaosNLI, introduced by Nie et al. (2020), is a large-scale dataset created to investigate human disagreement in natural language inference (NLI). It includes 100 annotations per example for 3,113 instances from SNLI and MNLI, as well as 1,532 examples from the $\alpha$NLI dataset, totaling around 464,500 annotations. In line with Javanmardi et al. (2024), we focus only on the SNLI and MNLI portions, which we refer to simply as ChaosNLI for convenience. Each entry provides rich metadata, including a unique identifier, the count of labels assigned by annotators, the majority label, the full label distribution, the distribution's entropy, the original text, and the original label from the source dataset. ChaosNLI facilitates detailed study of variability in human judgments, highlighting examples where disagreement is high and illustrating the limitations of treating the majority label as definitive ground truth. The dataset is publicly accessible under the *CC BY-NC 4.0 License*. For our experiments, we use the precomputed 768-dimensional embeddings, available at `https://github.com/alireza-javanmardi/conformal-credal-sets`, with further details on their generation provided by Javanmardi et al. (2024).

**QualityMRI**  Introduced by Obuchowicz et al. (2020), the QualityMRI dataset is part of the Data-Centric Image Classification (DCIC) Benchmark, which studies the role of dataset quality in shaping model performance. It consists of 310 magnetic resonance images that cover different quality levels, providing a resource for assessing MRI image quality. The dataset is distributed under the *Creative Commons BY-SA 4.0 License*.

**CIFAR-10**  CIFAR-10, introduced by Krizhevsky et al. (2009) and Geoffrey Hinton in 2009, is a widely adopted benchmark in machine learning and computer vision. It consists of 60,000 color

images with a resolution of $32 \times 32$ pixels, evenly divided among 10 classes: airplane, automobile, bird, cat, deer, dog, frog, horse, ship, and truck. The dataset is split into $50,000$ training images and $10,000$ test images, organized into five training batches and a single test batch, each containing $10,000$ images. CIFAR-10 is publicly available and has been extensively used for training and evaluating machine learning models. While the original dataset does not explicitly define a license, versions distributed through platforms such as TensorFlow datasets are provided under the *Creative Commons Attribution 4.0 License*.

**CIFAR-10H** CIFAR-10H provides human-generated soft labels for the $10,000$ images in the CIFAR-10 test set, reflecting the variability in human judgments for image classification. Introduced by Peterson et al. (2019), the dataset contains $511,400$ annotations from $2,571$ workers on Amazon Mechanical Turk, with each image receiving around 51 labels. Each annotation assigns an image to one of the ten CIFAR-10 classes, allowing the creation of a probability distribution over labels for every image. CIFAR-10H is publicly available under the *Creative Commons BY-NC-SA 4.0 License*.

**CIFAR-100** CIFAR-100, introduced by Krizhevsky et al. (2009), consists of $60,000$ color images at a resolution of $32 \times 32$ pixels, organized into 100 classes with 600 images per class. Each image carries both a "fine" label, indicating its specific class, and a "coarse" label corresponding to one of 20 broader superclasses. The dataset is divided into $50,000$ training images and $10,000$ test images. CIFAR-100 is derived from the Tiny Images dataset and is widely used for benchmarking image classification models. While the original dataset does not define a license, versions distributed through platforms such as TensorFlow Datasets are available under the *Creative Commons Attribution 4.0 License*.

**SVHN** The SVHN dataset, introduced by Netzer et al. (2011), contains over $600,000$ $32 \times 32$ RGB images of digits (0–9) extracted from real-world house numbers in Google Street View. It is organized into three subsets: $73,257$ images for training, $26,032$ for testing, and an additional set of $531,131$ images for extended training. SVHN is intended for digit recognition tasks and requires minimal preprocessing. Although the original dataset does not specify a license, versions distributed through platforms like TensorFlow Datasets are available under the *Creative Commons Attribution 4.0 License*.

**Places365** Places365, introduced by Zhou et al. (2018), is a large-scale dataset for scene recognition, comprising 1.8 million training images spanning 365 scene categories. The validation set contains 50 images per category, while the test set includes 900 images per category. An expanded variant, Places365-Challenge-2016, incorporates an additional 6.2 million images and 69 new scene categories, bringing the total to 8 million images across 434 categories. Although the original dataset does not specify a license, versions distributed through platforms such as TensorFlow Datasets are available under the *Creative Commons Attribution 4.0 License.*

**FMNIST** Fashion-MNIST (FMNIST), introduced by Xiao et al. (2017), contains $70,000$ grayscale images of Zalando products, each sized $28 \times 28$ pixels and categorized into 10 classes, including T-shirt/top, Trouser, and Sneaker. The dataset is divided into $60,000$ training images and $10,000$ test images, and it is commonly used as a modern replacement for the original MNIST dataset in machine learning benchmarks. *FMNIST is publicly released under the MIT License*.

**ImageNet** ImageNet, introduced by Deng et al. (2009), is a large-scale image dataset structured according to the WordNet hierarchy, comprising over 14 million images spanning more than $20,000$ categories. Its ILSVRC subset, commonly referred to as ImageNet-1K, contains $1,281,167$ training images, $50,000$ validation images, and $100,000$ test images across $1,000$ classes. *The dataset is freely accessible to researchers for non-commercial purposes*.

**TabArena Benchmark Data** TabArena (Erickson et al., 2025) is a continuously maintained benchmarking system designed for evaluating tabular machine learning models. It comprises 51 manually curated datasets representing real-world tabular tasks, including both classification and regression problems. Each dataset has been evaluated across 9 to 30 different splits, ensuring robust performance assessments. The datasets encompass a diverse range of domains, such as finance,

healthcare, and e-commerce, providing a comprehensive foundation for benchmarking various machine learning models. This diversity ensures that evaluations reflect the complexities and nuances found in real-world tabular data scenarios. We use 7 different datasets with IDs 46906, 46930, 46941, 46958, 46960, 46963, 46980 from the benchmark to validate our method in different experiments. Each of the datasets contains a classification task with 2 or more classes and a number of instances between $898$ and $12,684$. The datasets are publicly accessible and released under the *Apache 2.0 license*, ensuring permissive use and redistribution for research purposes.

### D.3 BASELINES

Below, we provide detailed descriptions of the baseline implementations used in this paper, relying on the implementations provided by Löhr et al. (2025).

**Credal Prediction based on Relative Likelihood (CreRL)**  The implementation of Credal Prediction based on Relative Likelihood (Löhr et al., 2025) is provided in `https://github.com/timoverse/credal-prediction-relative-likelihood`. We use the provided code to perform all experiments involving CreRL. Similar to our method, the CreRL defines plausibility in terms of the relative likelihood of a model. One significant difference is that, while CreRL tries to find sufficiently diverse hypotheses that satisfy the relative likelihood criterion, therefore having to train an ensemble of model, we directly obtain the plausible probability intervals by varying the logits of the maximum likelihood estimate.

**Credal Wrapper (CreWra)**  The Credal Wrapper (Wang et al., 2025a) was initially implemented in TensorFlow, but then reimplemented in PyTorch to ensure compatibility with other baselines. It follows a standard ensemble learning approach, training multiple models independently. Like our method, the Credal Wrapper constructs credal sets using class-wise upper and lower probability bounds, making it well-aligned with our implementation.

**Credal Ensembling (CreEns$_\alpha$)**  Our implementation adheres closely to the specifications outlined in Nguyen et al. (2025). The method extends standard ensemble training, adapting the inference stage by ranking predictions according to a distance metric and including only the top $\alpha\%$ of closest predictions when forming the credal sets. In our experiments, we employ the Euclidean distance and test multiple $\alpha$ values.

**Credal Deep Ensembles (CreNet)**  Since the official Credal Deep Ensembles implementation was provided only in TensorFlow, it was reimplemented by Löhr et al. (2025) in PyTorch to integrate seamlessly with other baselines. The version maintains the key design choices of the original, especially regarding the architecture and loss function. In particular, the models' final linear layers are replaced with a head that outputs $2 \times$ classes values corresponding to upper and lower probability bounds, followed by batch normalization and the custom IntSoftmax activation. The loss function applies standard cross-entropy to the upper bounds, while for the lower bounds, gradients are propagated only for the $\delta\%$ of samples exhibiting the largest errors, in line with Wang et al. (2025b). For our experiments, we adopt $\delta = 0.5$ as recommended in the original work.

**Credal Bayesian Deep Learning (CreBNN)**  The method proposed by Caprio et al. (2024a) was reimplemented by Löhr et al. (2025) using only the high-level description from the original work. The ensemble consists of Bayesian neural networks (BNNs), each trained via variational inference with distinct priors: the prior means $\mu$ are drawn from $[-1, 1]$ and the standard deviations $\sigma$ from $[0.1, 2]$, ensuring a diverse prior set. At inference time, we sample once from each BNN to generate a finite collection of probability distributions, and the credal set is defined as the convex hull of these samples.

**Evidential Deep Learning**  Our implementation of Evidential Deep Learning follows (Sensoy et al., 2018) as closely as possible. We use a single model and include a SoftPlus activation function after the last layer to ensure the output is non-negative. We use the Type II Maximum Likelihood as a loss function and the KL-divergence as a regularization term as described in (Sensoy et al., 2018). The regularization term is scaled by $\lambda_i = \min(1, i/10)$ at epoch $i$ as also done in the original work.

At inference time, the model predicts the evidence for each class, which can then be used to compute the parameters of the corresponding Dirichlet distribution.

**Deep Deterministic Uncertainty**   For the Deep Deterministic Uncertainty method (Mukhoti et al., 2023), we used the original implementation provided in `https://github.com/omegafragger/DDU`. This approach uses a single model to reason about uncertainty. In the original paper, the authors apply additional techniques—including spectral normalization and residual connections—to encourage more regularized embeddings in feature-space. For our comparison, we omit these techniques to ensure a fair comparison, as integrating such modifications into a pre-trained model would require re-training the model. Thus, we rely on the identical, trained `ResNet18`, which is also used for the other experiments. At inference time, epistemic uncertainty can be quantified through density estimation in feature-space: a normal distribution is fit to the embeddings of training data for each class, and epistemic uncertainty is computed on the basis of the likelihood of new embeddings under this distribution.

## D.4   COMPUTE RESOURCES

All experiments in this work were conducted using the computing resources listed in Table 2, with an estimated total GPU usage of approximately 820 hours.

**Table 2:** Specifications of Computing Resources

| Component | Specification |
|---|---|
| CPU | AMD EPYC MILAN 7413 Processor, 24C/48T 2.65GHz 128MB L3 Cache |
| GPU | 2 × NVIDIA A40 (48 GB GDDR each) |
| RAM | 128 GB (4x 32GB) DDR4-3200MHz ECC DIMM |
| Storage | 2 × 480GB Samsung Datacenter SSD PM893 |

## D.5   GENERATING SEMI-SYNTHETIC GROUND-TRUTH DISTRIBUTIONS

Due to a lack of ground-truth distributions, the evaluation of credal predictors remains non-trivial. While a number of datasets have a (test) set that includes multiple human annotations—such as the ones used in this work—most of the commonly-used benchmarking datasets do not provide these. Therefore, we use a simple method to generate semi-synthetic datasets that include (conditional) ground-truth distributions. The general idea is as follows: given a training set $\mathcal{D}_{\text{train}}$, we either train a model or retrieve a strong model from a model hub. The trained or retrieved model is then considered to be the ground-truth model $h^*$ and ground-truth distributions may be generated by collecting the predicted distributions $p(\cdot \mid \boldsymbol{x}, h^*)$ based on instances from the $\mathcal{D}_{\text{train}}$ or $\mathcal{D}_{\text{test}}$. The model that is to be evaluated (in terms of coverage and efficiency) is then trained on the same instances $\boldsymbol{x} \in \mathcal{D}_{\text{train}}$, but the labels are sampled from $p(\cdot \mid \boldsymbol{x}, h^*)$. Thereafter, the model can be evaluated using the test set.

For example, in Section 4.3, we train a `RandomForest` with the default parameters from scikit-learn (Pedregosa et al., 2011) with the exception of maximum depth; this is set to 5 to prevent the predicted distributions too "peaked". The `RandomForest` is assumed to be the ground-truth model $h^*$ and it's prediction for an instance $\boldsymbol{x}$ is taken to be the *ground-truth* conditional distribution $p(\cdot \mid \boldsymbol{x}, h^*)$. The `TabPFN` model is then trained on the same instances $\boldsymbol{x}$, but the labels $y$ are realizations sampled from the distribution $p(\cdot \mid \boldsymbol{x}, h^*)$. The model is then evaluated on the test set, for which the ground-truth distributions are also generated by the `RandomForest` $h^*$.

We refer to this as *semi-synthetic*, because, while the generated distribution is not (necessarily) the ground-truth, under the assumption that the used model is sufficiently well-trained, they should be "close" to the ground-truth.

## D.6   TURNING CLIP-BASED MODELS INTO ZERO-SHOT CLASSIFIERS

To demonstrate the usefulness and flexibility of our method for producing credal sets for any black-box model structure without the need for retraining, we apply it to multi-modal `CLIP`-based models.

Contrastive Language–Image Pretraining (`CLIP`) (Radford et al., 2021) introduced a mechanism to pre-train models that share embeddings across two modalities. The training data consists of a large corpus of images and their corresponding descriptions (e.g., captions or alternative text from websites). The central idea is to align each image with its textual description: images and their captions should be close in the embedding space, while mismatched pairs should be far apart. To achieve this, two modality-specific encoders are trained to produce embeddings of equal dimension, from which a similarity score (e.g., cosine similarity) is computed. Captions that accurately describe an image receive high similarity scores, whereas unrelated captions receive low scores. This training paradigm and model architecture have since been refined by subsequent works, yielding better-performing or more specialized models. For example, `BiomedCLIP` (Zhang et al., 2024), trained on biomedical data from `PubMed`, achieves superior performance on medical tasks. Similarly, the `SigLIP` (Zhai et al., 2023) and `SigLIP-2` (Tschannen et al., 2025) families adapt the training procedure and extend the datasets to include multilingual text sources, resulting in improved performance on general tasks (Zhai et al., 2023; Tschannen et al., 2025).

**Zero-Shot Prediction.** Zero-shot image classification with `CLIP`-based models proceeds by reformulating the label set into natural-language *templates*. For each candidate class, a short descriptive text is created (e.g., the template "a photo of a [label]" yields "a photo of a dog" or "a photo of a cat"). These textual descriptions are embedded by the text encoder, while the input image is embedded by the image encoder. The similarity between the image embedding and each text embedding is then computed, typically using cosine similarity. The resulting similarity values can be treated as logits, where the highest-scoring label determines the predicted class. Importantly, this formulation also makes it straightforward to restrict classification to any subset of labels without training a new classifier, since one can simply retain and compare the logits corresponding to the labels of interest. This procedure enables `CLIP`-based models to serve as flexible, task-agnostic classifiers without requiring any additional training, and has proven effective across diverse downstream domains (Radford et al., 2021; Zhang et al., 2024; Zhai et al., 2023; Tschannen et al., 2025).

**Templates for Multi-Lingual Datasets.** Zero-shot classification can be extended to multi-lingual datasets by translating labels into the target language and constructing corresponding templates. For example, in our experiments we used the English template "This is a photo of a [label]" alongside a Swahili template "Hii ni picha ya [label]", allowing classification in either language. Models such as `SigLIP-2` (Tschannen et al., 2025), trained on multilingual data, further improve robustness in this setting.

**Model Performance.** To illustrate the effectiveness of different `CLIP`-based models in our setting, we report their zero-shot classification accuracy on CIFAR-10, ImageNet, and DermMNIST (see Table 3). The results show that while standard `CLIP` performs strongly on general-purpose datasets, specialized variants such as `BiomedCLIP` yield improved performance on domain-specific tasks, and recent multilingual models like `SigLIP` and `SigLIP-2` further enhance accuracy on broad benchmarks.

**Table 3:** Zero-shot classification accuracy (%) of `CLIP`-based models on CIFAR-10 (EN = English, SW = Swahili, FR = French, ZH = Chinese), ImageNet, and DermaMNIST.

| Model | CIFAR-10 | | | | ImageNet | DermaMNIST |
|---|---|---|---|---|---|---|
| | **EN** | **SW** | **FR** | **ZH** | | |
| CLIP | 88.97% | 9.11% | 85.97% | 33.73% | 57.14% | 24.74% |
| SigLIP | 92.17% | **15.33%** | 84.05% | 91.28% | **72.83%** | 8.28% |
| SigLIP2 | **93.91%** | 10.21% | **92.21%** | **93.85%** | 69.87% | 11.67% |
| BiomedCLIP | – | – | – | – | – | **45.89%** |

# E  ADDITIONAL EXPERIMENTAL RESULTS

## E.1  COVERAGE VERSUS EFFICIENCY

In addition to the datasets provided in Section 4.1, we present an additional comparison to the baselines in the form of the QUALITYMRI dataset. Figure 8 shows that our approach Pareto dominates the

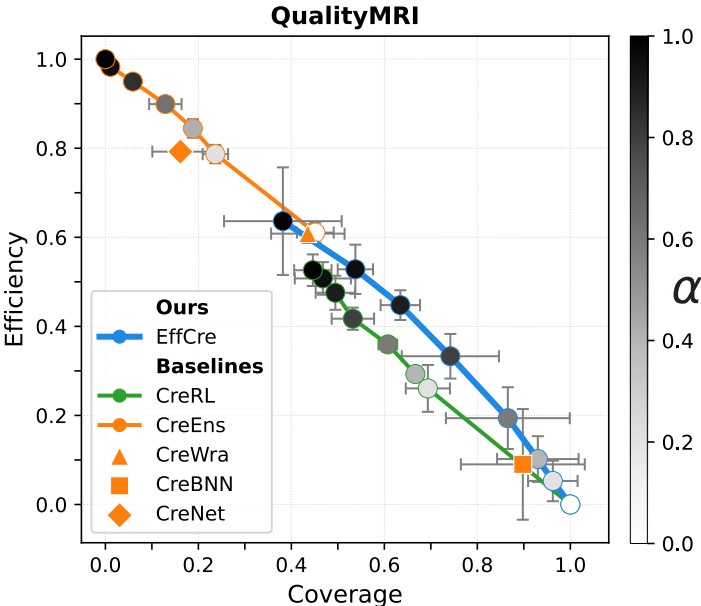

**Figure 8: Coverage versus Efficiency.**  Our method, EffCre, is compared to baselines on QUALITYMRI.

CreRL method, while having a similar coverage and efficiency to CreBNN for $\alpha = 0.4$. However, our method allows a trade-off between coverage and efficiency beyond that, allowing the exploration of regions with a better efficiency or better coverage. Our method is Pareto incomparable to the CreEns, because CreEns does not reach the high coverage region (while having higher efficiency, whereas our method does (while having lower efficiency). It should be noted that reaching the high coverage area, as our method does, is especially important in medical settings, as is the case for the QualityMRI dataset.

## E.2  OUT-OF-DISTRIBUTION DETECTION

For this experiment, we trained a `ResNet18` on CIFAR-10, which serves as the in-distribution dataset. At evaluation time, we consider both the in-distribution data and five out-of-distribution datasets to compute epistemic uncertainty values with our method. These values are then used to separate ID from OOD samples, with performance measured via AUROC. In addition to the results in Section 4.2, Table 5 reports AUROC scores across different $\alpha$ values for our method, and Table 4 states the corresponding training and inference

**Table 4:** Training and inference time in seconds for models trained on CIFAR10. Mean with standard deviation over three runs. Computed based on ensembles with 10 members.

| Method | Training time | Inference time |
|--------|---------------|----------------|
| EffCre | $2136.33 \pm 1.70$ | $1.50 \pm 0.02$ |
| CreRL | $12675.84 \pm 412.68$ | $11.46 \pm 0.12$ |
| CreWra | $21363.34 \pm 33.99$ | $11.44 \pm 0.19$ |
| CreEns | $21363.34 \pm 33.99$ | $11.44 \pm 0.23$ |
| CreNet | $24996.65 \pm 180.12$ | $11.41 \pm 0.18$ |
| CreBNN | $29796.67 \pm 12.94$ | $12.74 \pm 1.1$ |

times for each method. Furthermore, Appendix F.1 presents an ablation study on the effect of the ensemble size for OOD detection performance.

**Table 5:** Out-of-Distribution Detection.

| Method | SVHN | Places365 | CIFAR-100 | FMNIST | ImageNet |
|---|---|---|---|---|---|
| $\text{EffCre}_{0.0}$ | $0.478\pm0.006$ | $0.478\pm0.005$ | $0.480\pm0.005$ | $0.486\pm0.002$ | $0.481\pm0.002$ |
| $\text{EffCre}_{0.2}$ | $0.474\pm0.003$ | $0.474\pm0.001$ | $0.473\pm0.002$ | $0.474\pm0.002$ | $0.473\pm0.003$ |
| $\text{EffCre}_{0.4}$ | $0.303\pm0.100$ | $0.389\pm0.072$ | $0.335\pm0.040$ | $0.325\pm0.057$ | $0.338\pm0.035$ |
| $\text{EffCre}_{0.6}$ | $0.415\pm0.010$ | $0.428\pm0.007$ | $0.440\pm0.017$ | $0.404\pm0.024$ | $0.435\pm0.008$ |
| $\text{EffCre}_{0.8}$ | $0.744\pm0.009$ | $0.721\pm0.009$ | $0.720\pm0.007$ | $0.733\pm0.012$ | $0.700\pm0.008$ |
| $\text{EffCre}_{0.9}$ | $0.854\pm0.005$ | $0.827\pm0.006$ | $0.822\pm0.004$ | $0.860\pm0.005$ | $0.796\pm0.005$ |
| $\text{EffCre}_{0.95}$ | $0.885\pm0.003$ | $0.862\pm0.005$ | $0.854\pm0.003$ | $0.907\pm0.002$ | $0.826\pm0.004$ |
| $\text{EffCre}_{1.0}$ | $0.894\pm0.015$ | $0.886\pm0.008$ | $0.868\pm0.005$ | $0.933\pm0.010$ | $0.844\pm0.006$ |
| $\text{CreRL}_{0.95}$ | $0.917\pm0.013$ | $0.910\pm0.001$ | $0.901\pm0.000$ | $0.945\pm0.004$ | $0.878\pm0.002$ |
| CreWra | $0.957\pm0.003$ | $0.916\pm0.001$ | $0.916\pm0.000$ | $0.952\pm0.000$ | $0.890\pm0.001$ |
| $\text{CreEns}_{0.0}$ | $0.955\pm0.001$ | $0.913\pm0.000$ | $0.914\pm0.001$ | $0.949\pm0.001$ | $0.888\pm0.000$ |
| CreBNN | $0.907\pm0.006$ | $0.885\pm0.002$ | $0.880\pm0.002$ | $0.935\pm0.002$ | $0.859\pm0.002$ |
| CreNet | $0.943\pm0.003$ | $0.918\pm0.000$ | $0.912\pm0.000$ | $0.951\pm0.002$ | $0.884\pm0.001$ |

In the main paper, we focused exclusively on comparing credal predictors in order to ensure a consistent evaluation of methods within a single framework (that of credal predictors). This allows us to isolate the effect of the credal predictor from the influence of other factors such as the uncertainty measure or the base model. Here, we additionally compare our method to other methods that allow for uncertainty quantification with a single model. In particular, we compare EffCre to evidential deep learning (EDL) (Sensoy et al., 2018) and deep deterministic uncertainty (DDU) quantification (Mukhoti et al., 2023). For evidential deep learning, the epistemic uncertainty quantification is computed by

$$\text{EU} = \frac{K}{S},$$

where $K$ is the number of classes and $S = \sum_{k=1}^{K}(z_k + 1)$ denotes the sum of the predicted parameters of the Dirichlet distribution for an input for $x$. Deep deterministic uncertainty quantifies epistemic uncertainty on the basis of the likelihood of the embedding of an input

$$\text{EU} = \sum_{k=1}^{K} q(e \mid k)q(k),$$

where $q$ represents the density function of a normal distribution. The implementation details are described in Appendix D.3. The results are presented in Table 6. When compared to EDL, our

**Table 6:** Out-of-Distribution Detection.

| Method | SVHN | Places365 | CIFAR-100 | FMNIST | ImageNet |
|---|---|---|---|---|---|
| EDL | $0.938\pm0.010$ | $0.889\pm0.001$ | $0.887\pm0.001$ | $0.940\pm0.005$ | $0.866\pm0.001$ |
| DDU | $0.973\pm0.001$ | $0.969\pm0.001$ | $0.873\pm0.002$ | $0.892\pm0.010$ | $0.969\pm0.000$ |

method ($\text{EffCre}_{1.0}$) performs on par with EDL on Places365, while being slightly outperformed on the remaining datasets. However, it is important to emphasize that EDL requires re-training the model with a specific activation and loss function, hence it cannot be applied directly in standard settings, specifically if the training data is not available. In contrast, our method can be applied without re-training, making it compatible with a broader range of settings such as the ones using TabPFN and CLIP presented in the manuscript. Our method outperforms DDU on FMNIST, while being weaker on other datasets. Moreover, while DDU is also a post-training method, it requires access to the embeddings generated by the model, whereas our method does not. EffCre, by operating on logits, can be applied on top of black-box models, which enables it to be directly applied in more general settings, such as large language models, which form an interesting direction for future work. Overall, there is no clear winner across the evaluated methods, and drawing definitive conclusions remains challenging. Besides the stark differences in the working of the

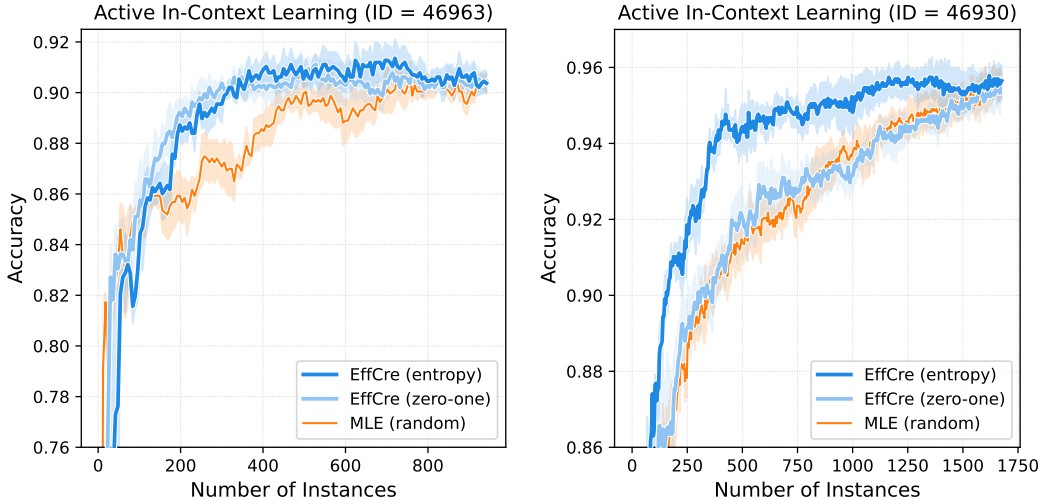

**Figure 9: Active In-Context Learning with `TabPFN`.** Performance on TabArena datasets 46963, and 46930 versus the random baseline.

methods, the OOD detection task itself comes with numerous caveats (Li et al., 2025), meaning that performance on this task can only be taken as a proxy of the quality of the epistemic uncertainty representation.

Additionally, although these methods also rely on a single (re-trained) model, they differ from the other (credal) approaches in that they do not produce credal sets. We consider this distinction to be particularly important, as the credal set predictors quantify a fundamentally different form of epistemic uncertainty than DDU. Indeed, the credal set represents an epistemic uncertainty with respect to the predicted probability distribution, which will directly affect the subsequent decision-making. DDU, however, quantifies a form of epistemic uncertainty about the "familiarity" of an input, derived from its density relative to the training data. It is not immediately clear how this should influence the decision-making process that follows.

### E.3   IN-CONTEXT LEARNING WITH TABPFN

We compute coverage and efficiency for our method used with `TabPFN` with all multi-class TABARENA datasets. As discussed in Section 4.3, the datasets do not come with ground-truth distributions. Therefore, we constuct *semi-synthethic* distributions that serve as the ground-truth. In Appendix D.5, we give a detailed explanation of our approach. Note that we consider the resulting distributions to only be a proxy of the "true" ground-truth distributions. In addition to computing the coverage and efficiency, we perform active in-context learning. In Section 4.3 and the results that will follow, this is done by first splitting the data, in a stratified manner, to have a 0.3 test split. The remaining 0.7 split is then split into an initial training set and the sampling pool, again stratified, such that the initial training set contains $2K$ instances, where $K$ is the number of classes. At every iteration, the predictor is "allowed" to sample $2K$ instances from the pool, based on its epistemic uncertainty, which is a common setup in active learning (Nguyen et al., 2019; Margraf et al., 2024). This is done until the pool is exhausted—and, hence, until the performance converges to what would be obtained with a traditional train-test split. The goal is thus, to select at every iteration samples that are most informative, i.e. the samples that will give the greatest performance increase at that iteration. In addition to the results in Section 4.3, we present active in-context learning results for two additional TABARENA datasets: 46963 and 46930. Figure 9 shows the results for our method using epistemic uncertainty sampling based on (5) and (6) compared to the random baseline. Conform the results presented in Section 4.3, our method applied to TABPFN provides a valuable advantage over the random baseline in terms of accuracy.

### E.4 ZERO-SHOT CLASSIFICATION WITH CLIP-BASED MODELS

Extending on the examples shown in Section 4.4, we create additional credal spider plots for `CLIP`-based models in Figures 10 to 12. Figure 11 highlights challenging natural images with high uncertainty in `CLIP`, Figure 12 examines medical images from DERMAMNIST, and Figure 10 analyzes cross-lingual predictions on CIFAR-10. Together, these visualizations complement the quantitative results reported in Table 3 by showcasing how credal spider plots reveal distinct uncertainty patterns that align with the models' performance across domains.

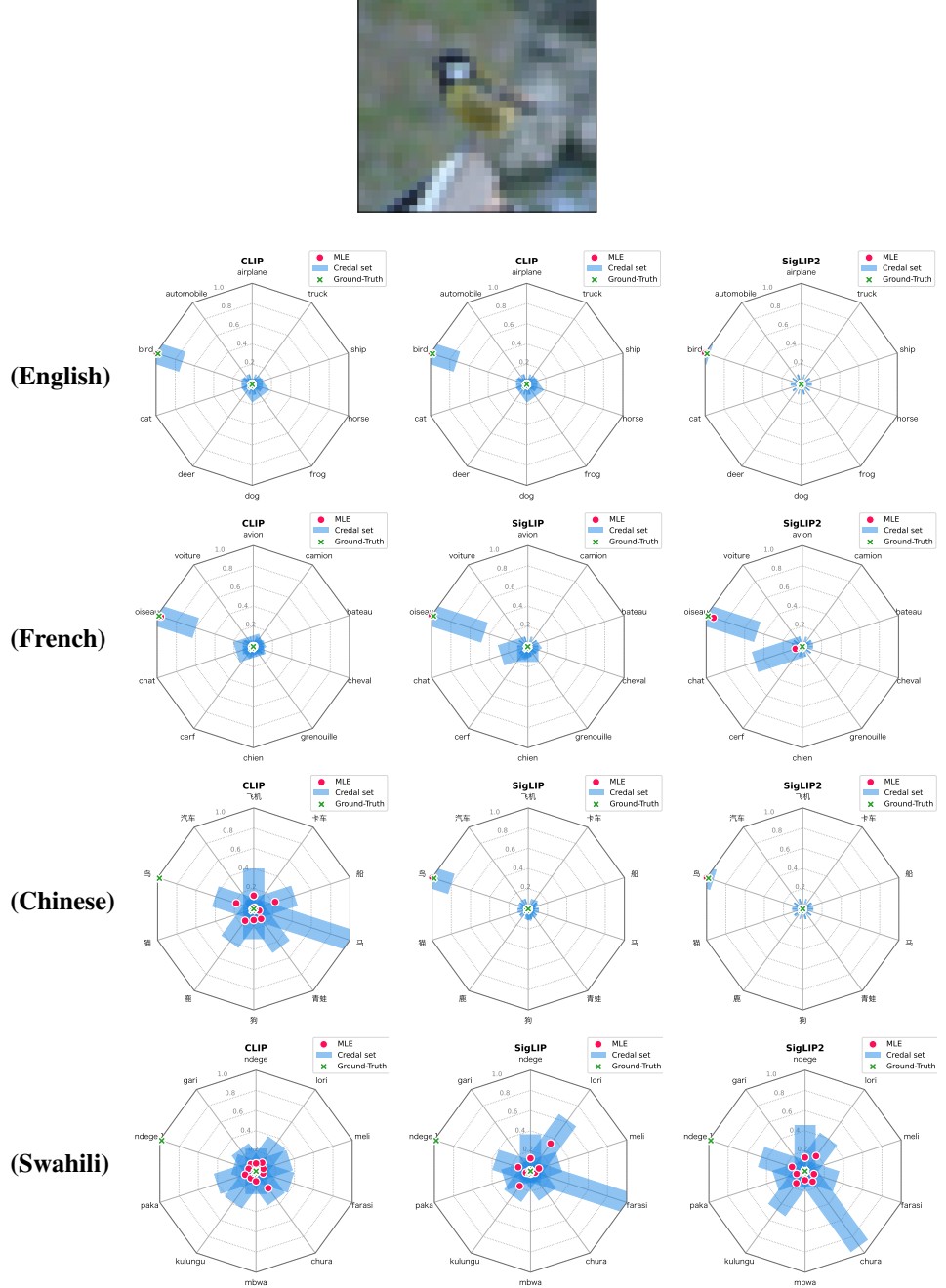

**Figure 10:** Credal spider plots for an image of a bird with `CLIP`, `SigLIP`, and `SigLIP-2` across different languages. In English, all models confidently predict the image as `bird`. In French, `SigLIP-2` maintains the correct maximum likelihood prediction but shows increased uncertainty toward `cat`. In Chinese, `CLIP` exhibits high uncertainty across all classes, indicating difficulties in this language, whereas `SigLIP` and `SigLIP-2` remain as confident as in English. In Swahili, all models struggle and display high uncertainty across all classes; notably, `bird` and `airplane` share the same word in Swahili, complicating the prediction. These examples align well with models' performances across the different languages (see Table 3).

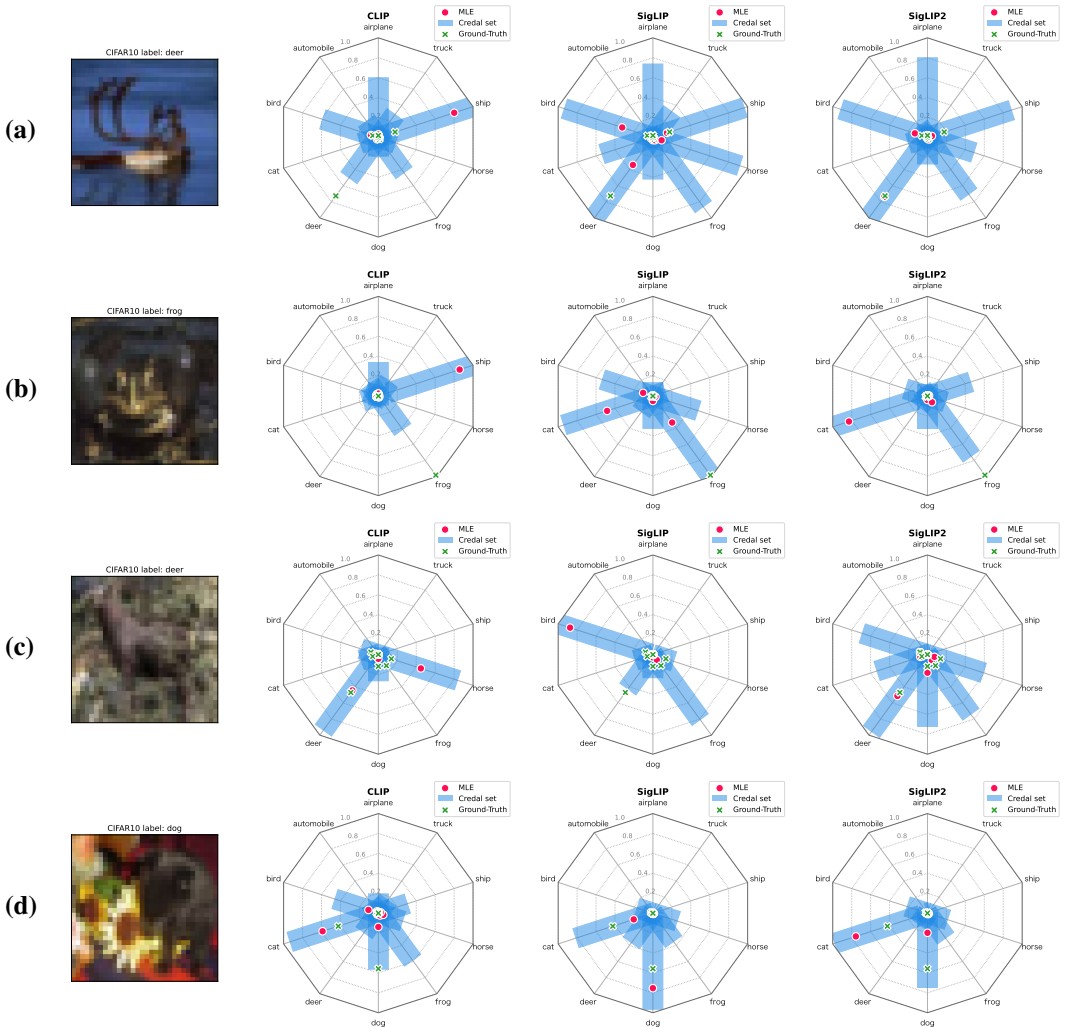

**Figure 11:** Comparison of credal sets for `CLIP`, `SigLIP`, and `SigLIP-2` on observations with high uncertainty with `CLIP`. Observation **(a)** shows a swimming deer, where the MLE is `ship`. High uncertainty is spread across `ship`, two sky-related classes (`airplane`, `bird`), the amphibious `frog`, and the correct class `deer`. Both `SigLIP` models exhibit similar patterns with even greater uncertainty. Observation **(b)** depicts a dark image of a frog misclassified as `ship`, with high uncertainty again on that class; both `SigLIP` models additionally assign probability to `cat`. Observation **(c)** is a challenging deer image, where annotators themselves showed high disagreement. `CLIP` is confident it is either `deer` or `horse`, while `SigLIP` favors `bird` or `frog`, and `SigLIP-2` remains certain it is an animal but not which. Observation **(d)** illustrates a case where human annotators are nearly evenly split between `cat` and `dog`, and the uncertainties of all three models capture this ambiguity.

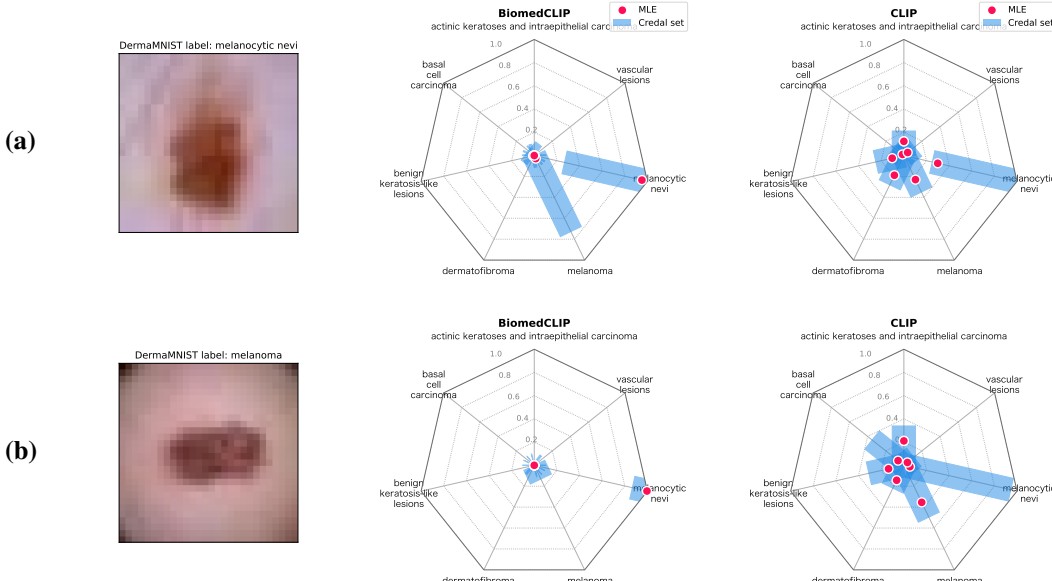

**Figure 12:** Comparison of credal sets for `BiomedCLIP` and `CLIP` on DERMAMNIST for a melanocytic nevi **(a)** and a melanoma **(b)**. While `BiomedCLIP` demonstrates higher overall performance than `CLIP` (see Table 3), it misclassifies the melanoma with high confidence and low uncertainty, which could be dangerous if applied in medical contexts. Interestingly, `CLIP` classifies the melanoma correctly, albeit with greater uncertainty. Both models predict the melanocytic nevi correctly, though `BiomedCLIP` shows increased uncertainty toward the related melanoma class, suggesting a more challenging case.

# F  ABLATIONS

This section contains additional ablation experiments.

## F.1  NUMBER OF ENSEMBLE MEMBERS IN OUT-OF-DISTRIBUTION DETECTION

We provide an additional ablation study on the impact of the ensemble size on out-of-distribution performance. Table 7 and Figure 13 demonstrate once more the efficiency of our approach: it requires only a single trained model. In contrast, ensemble-based baselines typically rely on at least five members and benefit from larger ensembles to improve performance.

**Table 7:** Ablation of different numbers of trained ensemble members for Out-of-Distribution Detection.

| Method | Members | SVHN | Places365 | CIFAR-100 | FMNIST | ImageNet |
|--------|---------|------|-----------|-----------|--------|----------|
| $\text{EffCre}_{0.95}$ | 1 | $0.885_{\pm 0.003}$ | $0.862_{\pm 0.005}$ | $0.854_{\pm 0.003}$ | $0.907_{\pm 0.002}$ | $0.826_{\pm 0.004}$ |
| $\text{CreRL}_{0.95}$ | 5 | $0.917_{\pm 0.012}$ | $0.894_{\pm 0.002}$ | $0.885_{\pm 0.002}$ | $0.928_{\pm 0.004}$ | $0.863_{\pm 0.002}$ |
| CreWra | 5 | $0.943_{\pm 0.006}$ | $0.904_{\pm 0.001}$ | $0.905_{\pm 0.001}$ | $0.939_{\pm 0.001}$ | $0.879_{\pm 0.001}$ |
| $\text{CreEns}_{0.0}$ | 5 | $0.938_{\pm 0.007}$ | $0.898_{\pm 0.001}$ | $0.900_{\pm 0.001}$ | $0.929_{\pm 0.001}$ | $0.874_{\pm 0.001}$ |
| CreBNN | 5 | $0.843_{\pm 0.006}$ | $0.829_{\pm 0.006}$ | $0.831_{\pm 0.007}$ | $0.851_{\pm 0.007}$ | $0.809_{\pm 0.007}$ |
| CreNet | 5 | $0.938_{\pm 0.003}$ | $0.908_{\pm 0.001}$ | $0.900_{\pm 0.001}$ | $0.941_{\pm 0.003}$ | $0.871_{\pm 0.002}$ |
| $\text{CreRL}_{0.95}$ | 10 | $0.921_{\pm 0.010}$ | $0.905_{\pm 0.002}$ | $0.896_{\pm 0.001}$ | $0.940_{\pm 0.002}$ | $0.872_{\pm 0.002}$ |
| CreWra | 10 | $0.953_{\pm 0.004}$ | $0.911_{\pm 0.001}$ | $0.912_{\pm 0.000}$ | $0.948_{\pm 0.001}$ | $0.886_{\pm 0.001}$ |
| $\text{CreEns}_{0.0}$ | 10 | $0.949_{\pm 0.001}$ | $0.907_{\pm 0.001}$ | $0.909_{\pm 0.002}$ | $0.941_{\pm 0.002}$ | $0.883_{\pm 0.001}$ |
| CreBNN | 10 | $0.880_{\pm 0.009}$ | $0.856_{\pm 0.002}$ | $0.859_{\pm 0.002}$ | $0.886_{\pm 0.001}$ | $0.838_{\pm 0.001}$ |
| CreNet | 10 | $0.944_{\pm 0.001}$ | $0.915_{\pm 0.001}$ | $0.908_{\pm 0.001}$ | $0.949_{\pm 0.001}$ | $0.881_{\pm 0.001}$ |
| $\text{CreRL}_{0.95}$ | 20 | $0.917_{\pm 0.013}$ | $0.910_{\pm 0.001}$ | $0.901_{\pm 0.000}$ | $0.945_{\pm 0.004}$ | $0.878_{\pm 0.002}$ |
| CreWra | 20 | $0.957_{\pm 0.003}$ | $0.916_{\pm 0.001}$ | $0.916_{\pm 0.000}$ | $0.952_{\pm 0.000}$ | $0.890_{\pm 0.001}$ |
| $\text{CreEns}_{0.0}$ | 20 | $0.955_{\pm 0.001}$ | $0.913_{\pm 0.000}$ | $0.914_{\pm 0.001}$ | $0.949_{\pm 0.001}$ | $0.888_{\pm 0.000}$ |
| CreBNN | 20 | $0.907_{\pm 0.006}$ | $0.885_{\pm 0.002}$ | $0.880_{\pm 0.002}$ | $0.935_{\pm 0.002}$ | $0.859_{\pm 0.002}$ |
| CreNet | 20 | $0.943_{\pm 0.003}$ | $0.918_{\pm 0.000}$ | $0.912_{\pm 0.000}$ | $0.951_{\pm 0.002}$ | $0.884_{\pm 0.001}$ |

## F.2  $\alpha$-VALUES FOR ACTIVE IN-CONTEXT LEARNING

We provide an additional ablation on the effect that the $\alpha$-value has on the performance of our method in active in-context learning. We evaluate runs for values $\alpha \in \{0.2, 0.4, 0.6, 0.8, 0.9, 0.95\}$. In Figure 14, we provide the results for the TabArena datasets with OpenML (Bischl et al., 2025) id 46941, 46963, and 46930. For the sake of legibility, we only consider the zero-one-loss-based epistemic uncertainty measure (6).

We observe that higher $\alpha$ values consistently improve performance across all three datasets until the performance converges at $\alpha = 0.8$. In particular, lower $\alpha$ values result in larger predicted sets with high epistemic uncertainty, which reduces the meaningful separation between instances. Consequently, the optimal order for selecting instances during active learning is lost when $\alpha$ is small, explaining the drop in performance.

## F.3  ACCURACY AND EXPECTED CALIBRATION SCORE EVALUATION FOR SINGLE MODELS

If we have to commit to a precise probabilistic prediction, a natural choice is to use the maximum likelihood estimate, which is a theoretically well-established approach. If, additionally, a class-wise prediction is sought, the argmax class can be predicted. To provide a sense of the quality of the underlying models trained for our experiments, we report standard supervised-learning metrics, namely accuracy and expected calibration error, for each individual model in Table 8 based on the original CIFAR-10 test set. This serves as a sanity check to ensure a fair comparison with the baselines. For our experiments with TabPFN and CLIP models we use the pre-trained models.

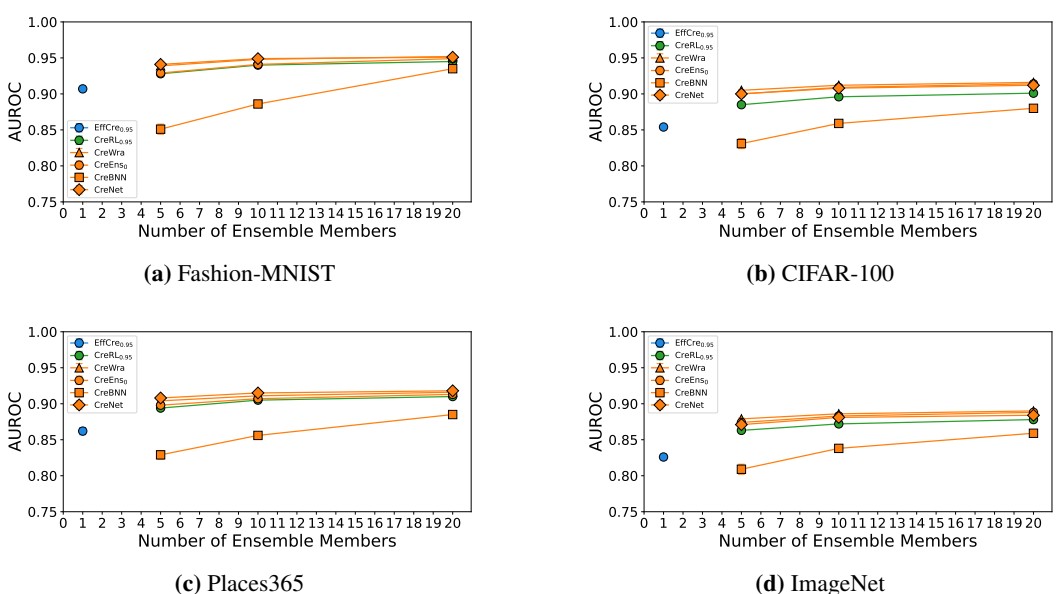

(a) Fashion-MNIST

(b) CIFAR-100

(c) Places365

(d) ImageNet

**Figure 13:** Out-of-distribution detection performance (based on AUROC score) as a function of ensemble size. CIFAR-10 is the in-distribution data while various datasets are used as OOD data.

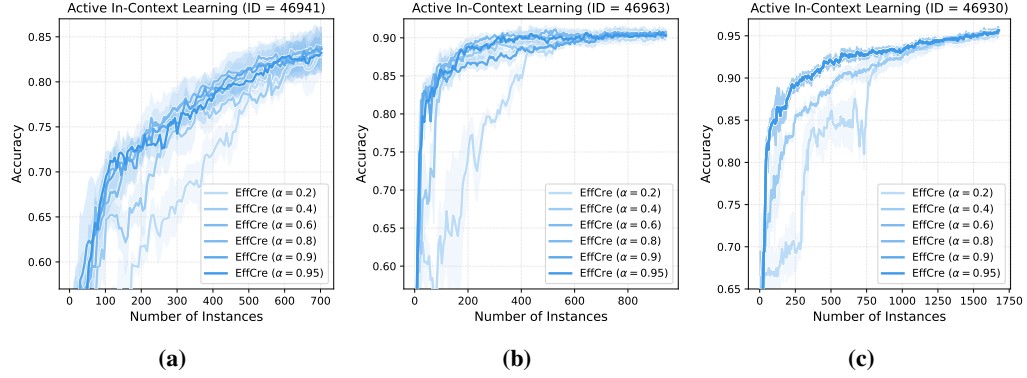

(a)

(b)

(c)

**Figure 14: Active In-Context Learning with `TabPFN`.** Performance on TabArena datasets 46941, 46963, 46930 for different values of $\alpha$.

| Method | Model | CIFAR10 | | ChaosNLI | | QualityMRI | |
| --- | --- | --- | --- | --- | --- | --- | --- |
| | | ECE | Acc | ECE | Acc | ECE | Acc |
| EffCre | 1 | $0.04 \pm 0.00$ | $0.93 \pm 0.00$ | $0.04 \pm 0.01$ | $0.61 \pm 0.01$ | $0.30 \pm 0.05$ | $0.49 \pm 0.05$ |
| CreRL$_{0.8}$ | 1 | $0.04 \pm 0.00$ | $0.94 \pm 0.00$ | $0.09 \pm 0.03$ | $0.60 \pm 0.02$ | $0.33 \pm 0.09$ | $0.48 \pm 0.04$ |
| | 2 | $0.06 \pm 0.01$ | $0.87 \pm 0.01$ | $0.13 \pm 0.01$ | $0.57 \pm 0.01$ | $0.35 \pm 0.08$ | $0.50 \pm 0.05$ |
| | 3 | $0.06 \pm 0.00$ | $0.87 \pm 0.00$ | $0.10 \pm 0.01$ | $0.56 \pm 0.01$ | $0.37 \pm 0.10$ | $0.51 \pm 0.02$ |
| | 4 | $0.06 \pm 0.00$ | $0.88 \pm 0.00$ | $0.09 \pm 0.01$ | $0.58 \pm 0.01$ | $0.30 \pm 0.09$ | $0.49 \pm 0.03$ |
| | 5 | $0.06 \pm 0.00$ | $0.89 \pm 0.00$ | $0.12 \pm 0.01$ | $0.60 \pm 0.00$ | $0.38 \pm 0.09$ | $0.48 \pm 0.04$ |
| | 6 | $0.06 \pm 0.00$ | $0.89 \pm 0.00$ | $0.10 \pm 0.01$ | $0.57 \pm 0.01$ | $0.30 \pm 0.08$ | $0.49 \pm 0.07$ |
| | 7 | $0.06 \pm 0.01$ | $0.88 \pm 0.01$ | $0.09 \pm 0.01$ | $0.59 \pm 0.02$ | $0.34 \pm 0.04$ | $0.51 \pm 0.04$ |
| | 8 | $0.06 \pm 0.01$ | $0.90 \pm 0.00$ | $0.11 \pm 0.01$ | $0.60 \pm 0.01$ | $0.33 \pm 0.09$ | $0.49 \pm 0.04$ |
| | 9 | $0.06 \pm 0.00$ | $0.90 \pm 0.01$ | $0.09 \pm 0.01$ | $0.59 \pm 0.00$ | $0.35 \pm 0.10$ | $0.47 \pm 0.03$ |
| | 10 | $0.06 \pm 0.00$ | $0.91 \pm 0.00$ | $0.09 \pm 0.01$ | $0.39 \pm 0.01$ | $0.38 \pm 0.09$ | $0.49 \pm 0.04$ |
| CreWra | 1 | $0.03 \pm 0.00$ | $0.94 \pm 0.00$ | $0.12 \pm 0.02$ | $0.60 \pm 0.01$ | $0.39 \pm 0.02$ | $0.48 \pm 0.03$ |
| | 2 | $0.03 \pm 0.00$ | $0.95 \pm 0.00$ | $0.11 \pm 0.01$ | $0.60 \pm 0.01$ | $0.39 \pm 0.02$ | $0.51 \pm 0.04$ |
| | 3 | $0.03 \pm 0.00$ | $0.94 \pm 0.00$ | $0.12 \pm 0.02$ | $0.59 \pm 0.01$ | $0.38 \pm 0.02$ | $0.48 \pm 0.03$ |
| | 4 | $0.03 \pm 0.00$ | $0.94 \pm 0.00$ | $0.15 \pm 0.01$ | $0.57 \pm 0.02$ | $0.34 \pm 0.01$ | $0.49 \pm 0.02$ |
| | 5 | $0.03 \pm 0.00$ | $0.94 \pm 0.00$ | $0.13 \pm 0.01$ | $0.55 \pm 0.01$ | $0.37 \pm 0.03$ | $0.46 \pm 0.01$ |
| | 6 | $0.03 \pm 0.00$ | $0.94 \pm 0.00$ | $0.10 \pm 0.01$ | $0.59 \pm 0.02$ | $0.38 \pm 0.04$ | $0.48 \pm 0.04$ |
| | 7 | $0.03 \pm 0.00$ | $0.94 \pm 0.00$ | $0.12 \pm 0.01$ | $0.59 \pm 0.01$ | $0.39 \pm 0.02$ | $0.46 \pm 0.05$ |
| | 8 | $0.03 \pm 0.00$ | $0.94 \pm 0.00$ | $0.13 \pm 0.01$ | $0.59 \pm 0.01$ | $0.34 \pm 0.01$ | $0.51 \pm 0.06$ |
| | 9 | $0.03 \pm 0.00$ | $0.94 \pm 0.00$ | $0.12 \pm 0.02$ | $0.58 \pm 0.01$ | $0.33 \pm 0.02$ | $0.48 \pm 0.04$ |
| | 10 | $0.03 \pm 0.00$ | $0.94 \pm 0.00$ | $0.10 \pm 0.01$ | $0.58 \pm 0.02$ | $0.34 \pm 0.04$ | $0.47 \pm 0.04$ |
| CreEns | 1 | $0.03 \pm 0.00$ | $0.94 \pm 0.00$ | $0.12 \pm 0.02$ | $0.60 \pm 0.02$ | $0.34 \pm 0.04$ | $0.47 \pm 0.04$ |
| | 2 | $0.03 \pm 0.00$ | $0.94 \pm 0.00$ | $0.10 \pm 0.02$ | $0.60 \pm 0.01$ | $0.32 \pm 0.01$ | $0.48 \pm 0.02$ |
| | 3 | $0.03 \pm 0.00$ | $0.94 \pm 0.00$ | $0.11 \pm 0.02$ | $0.60 \pm 0.00$ | $0.34 \pm 0.04$ | $0.47 \pm 0.04$ |
| | 4 | $0.03 \pm 0.00$ | $0.94 \pm 0.00$ | $0.14 \pm 0.01$ | $0.57 \pm 0.02$ | $0.38 \pm 0.04$ | $0.48 \pm 0.04$ |
| | 5 | $0.03 \pm 0.00$ | $0.94 \pm 0.00$ | $0.07 \pm 0.03$ | $0.59 \pm 0.08$ | $0.31 \pm 0.02$ | $0.46 \pm 0.03$ |
| | 6 | $0.03 \pm 0.00$ | $0.94 \pm 0.00$ | $0.12 \pm 0.03$ | $0.58 \pm 0.01$ | $0.33 \pm 0.02$ | $0.48 \pm 0.04$ |
| | 7 | $0.03 \pm 0.00$ | $0.94 \pm 0.00$ | $0.09 \pm 0.03$ | $0.59 \pm 0.01$ | $0.34 \pm 0.01$ | $0.49 \pm 0.02$ |
| | 8 | $0.03 \pm 0.00$ | $0.94 \pm 0.00$ | $0.12 \pm 0.03$ | $0.59 \pm 0.01$ | $0.39 \pm 0.02$ | $0.51 \pm 0.04$ |
| | 9 | $0.03 \pm 0.00$ | $0.94 \pm 0.00$ | $0.10 \pm 0.01$ | $0.59 \pm 0.01$ | $0.37 \pm 0.02$ | $0.47 \pm 0.03$ |
| | 10 | $0.03 \pm 0.00$ | $0.94 \pm 0.00$ | $0.11 \pm 0.04$ | $0.59 \pm 0.01$ | $0.38 \pm 0.04$ | $0.48 \pm 0.04$ |
| CreNet | 1 | $0.04 \pm 0.01$ | $0.93 \pm 0.00$ | $0.11 \pm 0.02$ | $0.59 \pm 0.01$ | $0.41 \pm 0.02$ | $0.47 \pm 0.02$ |
| | 2 | $0.03 \pm 0.00$ | $0.95 \pm 0.00$ | $0.11 \pm 0.02$ | $0.59 \pm 0.01$ | $0.39 \pm 0.02$ | $0.50 \pm 0.04$ |
| | 3 | $0.02 \pm 0.00$ | $0.95 \pm 0.01$ | $0.11 \pm 0.02$ | $0.59 \pm 0.01$ | $0.38 \pm 0.03$ | $0.48 \pm 0.03$ |
| | 4 | $0.03 \pm 0.00$ | $0.94 \pm 0.00$ | $0.15 \pm 0.01$ | $0.57 \pm 0.02$ | $0.34 \pm 0.01$ | $0.49 \pm 0.02$ |
| | 5 | $0.03 \pm 0.00$ | $0.93 \pm 0.00$ | $0.13 \pm 0.01$ | $0.55 \pm 0.01$ | $0.37 \pm 0.03$ | $0.46 \pm 0.01$ |
| | 6 | $0.03 \pm 0.00$ | $0.94 \pm 0.00$ | $0.10 \pm 0.01$ | $0.57 \pm 0.02$ | $0.38 \pm 0.04$ | $0.48 \pm 0.05$ |
| | 7 | $0.03 \pm 0.00$ | $0.92 \pm 0.01$ | $0.12 \pm 0.01$ | $0.59 \pm 0.01$ | $0.39 \pm 0.02$ | $0.47 \pm 0.03$ |
| | 8 | $0.02 \pm 0.01$ | $0.94 \pm 0.00$ | $0.13 \pm 0.01$ | $0.59 \pm 0.00$ | $0.34 \pm 0.01$ | $0.51 \pm 0.06$ |
| | 9 | $0.03 \pm 0.00$ | $0.94 \pm 0.02$ | $0.12 \pm 0.02$ | $0.58 \pm 0.01$ | $0.33 \pm 0.02$ | $0.48 \pm 0.04$ |
| | 10 | $0.03 \pm 0.01$ | $0.94 \pm 0.00$ | $0.10 \pm 0.01$ | $0.58 \pm 0.02$ | $0.34 \pm 0.04$ | $0.48 \pm 0.03$ |
| CreBNN | 1 | $0.64 \pm 0.00$ | $0.87 \pm 0.00$ | $0.10 \pm 0.04$ | $0.49 \pm 0.07$ | $0.15 \pm 0.14$ | $0.54 \pm 0.11$ |
| | 2 | $0.64 \pm 0.01$ | $0.87 \pm 0.02$ | $0.09 \pm 0.03$ | $0.49 \pm 0.07$ | $0.15 \pm 0.14$ | $0.54 \pm 0.11$ |
| | 3 | $0.65 \pm 0.00$ | $0.88 \pm 0.01$ | $0.10 \pm 0.05$ | $0.49 \pm 0.08$ | $0.15 \pm 0.04$ | $0.54 \pm 0.11$ |
| | 4 | $0.65 \pm 0.01$ | $0.88 \pm 0.01$ | $0.07 \pm 0.00$ | $0.44 \pm 0.00$ | $0.17 \pm 0.12$ | $0.54 \pm 0.11$ |
| | 5 | $0.65 \pm 0.00$ | $0.88 \pm 0.00$ | $0.07 \pm 0.00$ | $0.44 \pm 0.00$ | $0.24 \pm 0.13$ | $0.54 \pm 0.12$ |
| | 6 | $0.65 \pm 0.01$ | $0.88 \pm 0.01$ | $0.13 \pm 0.05$ | $0.55 \pm 0.08$ | $0.28 \pm 0.14$ | $0.46 \pm 0.11$ |
| | 7 | $0.64 \pm 0.00$ | $0.87 \pm 0.00$ | $0.11 \pm 0.03$ | $0.53 \pm 0.06$ | $0.14 \pm 0.15$ | $0.44 \pm 0.09$ |
| | 8 | $0.63 \pm 0.01$ | $0.86 \pm 0.02$ | $0.13 \pm 0.04$ | $0.55 \pm 0.07$ | $0.19 \pm 0.07$ | $0.53 \pm 0.13$ |
| | 9 | $0.63 \pm 0.01$ | $0.85 \pm 0.04$ | $0.12 \pm 0.05$ | $0.53 \pm 0.07$ | $0.19 \pm 0.07$ | $0.51 \pm 0.11$ |
| | 10 | $0.63 \pm 0.00$ | $0.86 \pm 0.01$ | $0.12 \pm 0.04$ | $0.54 \pm 0.07$ | $0.17 \pm 0.12$ | $0.52 \pm 0.12$ |

**Table 8:** Comparison of ECE and accuracy of single models per method across datasets.

