# OpenReview forum: "Efficient Credal Prediction through Decalibration"
_ICLR.cc/2026/Conference — ICLR 2026 Poster_

### Official Review · Reviewer_9AwT · 2025-10-28

**Soundness:** 2
**Presentation:** 2
**Contribution:** 2
**Rating:** 4
**Confidence:** 4

**Summary:**

In this work, the authors present a post-hoc approach for generating the credal prediction by using class-wise plausible probability intervals. This is achieved by perturbing a trained model’s logits under a global likelihood-ratio budget, thereby exploring less-likely yet still plausible predictions without retraining. Multiple experiments from different perspectives are conducted.

**Strengths:**

1. The paper is relatively well-structured and easy to follow.

2. Although the work is built on recent work--the likelihood-based notion of plausibility (Löhr et al., 2025), the motivation and technical routes are different. It is novel and interesting to me.

3. Mathematical proofs for the relative propositions are provided.

4. Multiple experiments are performed.

**Weaknesses:**

1. *Extensive experiments show that our method yields credal sets with strong coverage and efficiency and performs well on out-of-distribution detection tasks.* The main empirical claim seems a bit misleading. As from the OOD detection benchmarks, e.g., in Table 5, the EffEct only performs reasonably when \alpha is close to 1, e.g., $\alpha$ = 0.95 (still visibly lower than the other baselines). If we fix to use these values of $\alpha$, how would one conclude that the EffCre has strong coverage and efficiency?

2. The practicality of evaluating efficiency and coverage is limited. A key difficulty in supervised learning is the absence of ground truth for test instances. In addition, this work lacks theoretical guarantees for the coverage and does not provide a clear recipe for choosing the parameter $\alpha$.

3. In classification tasks, the prediction performance, e.g., test accuracy and calibration performance (expected calibration error), also matters. The performance of this approach in this matter remains unclear. As well as how $\alpha$ will influnce the prediction performance.

4. The paper highlights its good performance in epistemic uncertainty estimation for OOD detection using a single model. To support this claim, it would be valuable to include comparisons with other single-model-based epistemic uncertainty estimation methods, such as those from the evidential deep learning family or deterministic approaches. If EffCre continues to outperform these additional baselines, it would substantially strengthen the paper’s significance and credibility.

**Questions:**

1. How would the authors place the work? A theoretical one or practical work? What would be the potential practical use case for the approach?

2. Proposition 2.1 is a desired design principle for this method, right? It is a sufficient condition for controlling the trade-off between efficiency and coverage, not a necessary condition, am I correct?

3. The EffCre significantly reduces the training complexity via a single model. What is the inference time complexity of EffCre?

---

> ### Author Response · Authors · 2025-11-20
> **Response to Reviewer 9AwT (1)**
>
> We thank you for carefully reading our work and providing valuable feedback. We appreciate that you recognize the novelty of our approach and acknowledge the strength of its theoretical foundation. We address your remaining questions and comments below. We have additionally uploaded an updated version of the paper containing the requested changes in blue.
>
> **W1:** *“The main empirical claim seems a bit misleading. As from the OOD detection benchmarks, e.g., in Table 5, the EffEct only performs reasonably when \alpha is close to 1, e.g., *$\alpha$* = 0.95 (still visibly lower than the other baselines). If we fix to use these values of *$\alpha$*, how would one conclude that the EffCre has strong coverage and efficiency?”*
>
> We would like to clarify that evaluating the coverage-efficiency trade-off and OOD detection performance are two distinct objectives. Coverage and efficiency inherently involve a trade-off, which in our approach can be balanced by adjusting $\alpha$. Consequently, there is no single setting that simultaneously maximizes both, and the optimal choice of $\alpha$ will also depend on the specific benchmark task at hand. Importantly, our experiments demonstrate that EffCre achieves strong performance with respect to the coverage–efficiency trade-off compared to state-of-the-art baselines. On the other hand, as you rightfully point out, for OOD detection, a value of $\alpha$ closer to 1 is beneficial, as it produces very efficient (small) sets for in-distribution data, thereby improving separation from out-of-distribution samples.
>
> We consider this adaptability to be a key advantage of our approach: one can adjust $\alpha$ to suit the task at hand. Having said that, we acknowledge that the original statement could be misleading, so we have updated it to clearly distinguish the two tasks and their respective performance metrics.
>
> **W2:** *“The practicality of evaluating efficiency and coverage is limited. A key difficulty in supervised learning is the absence of ground truth for test instances. In addition, this work lacks theoretical guarantees for the coverage and does not provide a clear recipe for choosing the parameter.”*
>
> We consider this criticism to address a general challenge in uncertainty evaluation, rather than a limitation specific to our approach. In supervised learning, the absence of ground-truth distributions is a well-known and inherent limitation shared by all methods in the field. To address this evaluation challenge, we incorporate a semi-synthetic setup (Section 4.3, Appendix D.5) in which ground-truth distributions, while not directly observable, can be reliably approximated through an auxiliary model. Besides this, to the best of our knowledge, there are no other methods that can provide guarantees on the coverage of credal sets without access to ground-truth distributions. While conformal prediction-style methods can give such guarantees, they require ground-truth distributions for calibration.
>
> **W3:** *“In classification tasks, the prediction performance, e.g., test accuracy and calibration performance (expected calibration error), also matters. The performance of this approach in this matter remains unclear. As well as how  will influnce the prediction performance.”*
>
> If we have to commit to a precise probabilistic prediction, a natural choice is to use the maximum likelihood estimate, which is a theoretically well-established approach. If, additionally, a class-wise prediction is sought, the argmax class can be predicted. Our decalibration approach relies on the MLE, but does not alter it, therefore we do not incur any performance decrease.
>
> That said, we have added standard accuracy scores and expected calibration errors for the model used for our method and for the models used for the baselines in Appendix F.3. All metrics are computed on the original CIFAR-10 test set.

---

> ### Author Response · Authors · 2025-11-20
> **Response to Reviewer 9AwT (2)**
>
> **W4:** *“The paper highlights its good performance in epistemic uncertainty estimation for OOD detection using a single model. To support this claim, it would be valuable to include comparisons with other single-model-based epistemic uncertainty estimation methods, such as those from the evidential deep learning family or deterministic approaches. If EffCre continues to outperform these additional baselines, it would substantially strengthen the paper’s significance and credibility.”*
>
> We focused exclusively on comparing credal predictors in the submitted version of this work in order to ensure a consistent evaluation of methods within a single framework (that of credal predictors). This allows us to isolate the effect of the credal predictor from the influence of other factors such as the uncertainty measure or the base model.
> Regardless, we want to thank you for pointing us to these baselines! We now include additional OoD detection results for the baseline methods Evidential Deep Learning (EDL) [1] and Deep Deterministic Uncertainty (DDU) [2].
>
> | Method      | SVHN            | Places365        | CIFAR-100        | FMNIST          | ImageNet        |
> |-------------|------------------|------------------|------------------|------------------|------------------|
> | EffCre$_{1.0}$  | 0.894 ± 0.015    | 0.886 ± 0.008    | 0.868 ± 0.005    | 0.933 ± 0.010    | 0.844 ± 0.006    |
> | EDL         | 0.938 ± 0.010    | 0.889 ± 0.001    | 0.887 ± 0.001    | 0.940 ± 0.005    | 0.866 ± 0.001    |
> | DDU         | 0.973 ± 0.001    | 0.969 ± 0.001    | 0.873 ± 0.002    | 0.892 ± 0.010    | 0.969 ± 0.000    |
>
> When compared to EDL, our method (EffCre) performs on par with EDL on Places365, while being slightly outperformed on the remaining datasets. However, it is important to emphasize that EDL requires re-training the model with a specific activation and loss function, hence it cannot be applied directly in standard settings, specifically if the training data is not available. In contrast, our method can be applied without re-training, making it compatible with a broader range of settings such as the ones using TabPFN and CLIP presented in the manuscript.
>
> Our method outperforms DDU on FMNIST, while being weaker on other datasets. Moreover, while DDU is also a post-training method, it requires access to the embeddings generated by the model, whereas our method does not. EffCre, by operating on logits, can be applied on top of black-box models, which enables it to be directly applied in more general settings, such as large language models, which form an interesting direction for future work. Overall, there is no clear winner across the evaluated methods, and drawing definitive conclusions remains challenging. Besides the stark differences in the working of the methods, the OOD detection task itself comes with numerous caveats [3], meaning that performance on this task can only be taken as a proxy of the quality of the epistemic uncertainty representation.
>
> Currently, we have added the additional comparison table to Appendix E.2 (Table 6) to keep it separate from the comparison with credal predictors. This preserves a smooth reading flow in the main paper and clarifies that, although these newly considered methods also rely on a single (re-trained) model, they differ from the other approaches in that they do not produce credal sets. We consider this distinction to be particularly important, as the credal set predictors quantify a fundamentally different form of epistemic uncertainty than DDU. Indeed, the credal set represents an epistemic uncertainty with respect to the predicted probability distribution, which will directly affect the subsequent decision-making. DDU, however, quantifies a form of epistemic uncertainty about the “familiarity” of an input, derived from its density relative to the training data. It is not immediately clear how this should influence the decision-making process that follows.
>
> We would appreciate your feedback on the presented additional baseline comparisons. Specifically: 1. Are you satisfied with these comparisons or did you have any other particular methods in mind? 2. Do you find the distinction between credal predictors and the other approaches to be meaningful and appropriate in this context?
>
> [1] Sensoy et al. Evidential Deep Learning to Quantify Classification Uncertainty. 2018.
>
> [2] Mukhoti et al. Deep Deterministic Uncertainty: A New Simple Baseline. 2023.
>
> [3] Li et al. Out-of-Distribution Detection Methods Answer the Wrong Questions. 2025.

---

> ### Author Response · Authors · 2025-11-20
> **Response to Reviewer 9AwT (3)**
>
> **Q1:** *“How would the authors place the work? A theoretical one or practical work? What would be the potential practical use case for the approach?”*
>
> We believe our contributions to be twofold. First, we have formalized the problem and given a theoretical analysis of the solution proposed by us. Second, we demonstrate the empirical effectiveness of our method using real-world, benchmark datasets and models. For example, in Section 4.3, we evaluate our method using TabPFN and (amongst others) the medical, tabular dataset MIC (ID: 46980), which serves as an illustrative example of a practical use case. In medical settings, there is usually a limited amount of labelled data, because annotation by clinicians is costly. In such cases, practitioners may use our method to leverage the strongest available models (e.g. foundation-style models) while minimizing labelling effort using active in-context learning, as demonstrated. In general, our method could be used in practical settings where the user benefits from using pre-trained models that may otherwise not be able to provide (epistemic) uncertainty.
>
> **Q2:** *“Proposition 2.1 is a desired design principle for this method, right? It is a sufficient condition for controlling the trade-off between efficiency and coverage, not a necessary condition, am I correct?”*
>
> Proposition 2.1 captures a desired design principle: the lower and upper probability bounds, and therefore the coverage implied by our set construction, increase monotonically in $\alpha$. Indeed, this is a sufficient condition, which follows naturally from using the relative likelihood-based approach. However, it is not a necessary condition, as alternative construction principles can also yield such nested credal sets. For example, one could train an ensemble of models with additional constraints on the lower and upper probability intervals of each class. However, we opt for a likelihood-based approach as it enjoys well-established theoretical properties, including parameterization invariance, and principled inference without the need to specify a prior.
>
> **Q3:** *“The EffCre significantly reduces the training complexity via a single model. What is the inference time complexity of EffCre?”*
>
> The inference complexity of EffCre is very low. Our method requires only a single forward pass through the model to obtain the logits, after which we simply add a constant to these logits and apply a softmax operation. This introduces negligible computational overhead, resulting in very fast inference.
>
> While we had already reported the training times in Table 4, we have now added the corresponding inference times as well (see the table below, also included in updated Table 4 in the paper). The results clearly confirm the efficiency of our approach: EffCre achieves inference times that are orders of magnitude lower than all baselines.
>
> | Method  | Training time (s)         | Inference time (s)        |
> |---------|----------------------------|----------------------------|
> | EffCre  | 2136.33 ± 1.70             | 1.50 ± 0.02                |
> | CreRL   | 12675.84 ± 412.68          | 11.46 ± 0.12               |
> | CreWra  | 21363.34 ± 33.99           | 11.44 ± 0.19               |
> | CreEns  | 21363.34 ± 33.99           | 11.44 ± 0.23               |
> | CreNet  | 24996.65 ± 180.12          | 11.41 ± 0.18               |
> | CreBNN  | 29796.67 ± 12.94           | 12.74 ± 1.10               |

---

> ### Comment · Reviewer_9AwT · 2025-11-28
>
> I thank the authors for their detailed responses and new results in the rebuttal.
>
> 1. I believe that providing the efficiency and coverage of the predicted credal set efficiently is a valuable contribution of this work, which I hope could trigger much more work to be done in this community to enhance the practical impact of the work further.
>
> 2. The added statement revisions improve the clarity of the paper.
>
> 3. The new results of the inference and training time further support the main claim of efficiency.
>
> 4. The ECE comparisons with *single models* show that the ECE performance is comparable. (It might also be logical to  compare with the real 'ensemble' version of the credal predictors (I am not asking for additional experiments.))
>
> 5. *When compared to EDL, our method (EffCre) performs on par with EDL on Places365, while being slightly outperformed on the remaining datasets.* When I read the reported numbers in the table, I can not draw this conclusion. Instead, the EDL outperforms the EffCre. In terms of aiming for better OOD performance using a single model, DDU and EDL seem to be better choices, especially since the original DDU paper showed its strong generalization ability in larger datasets and different neural networks. I also agree with the authors' point that EffCre's advantages are training-free.
>
> 6. How to get the optimal $\alpha$ per task (a more interpretable and practical recipe of tuning $\alpha$) seems to be a difficult but definitely useful future work.
>
> Therefore, I would like to increase my score to 6 (If the rating system allows, I would give a score of 7).

---

### Official Review · Reviewer_XAJU · 2025-11-01

**Soundness:** 3
**Presentation:** 3
**Contribution:** 4
**Rating:** 8
**Confidence:** 3

**Summary:**

The paper tackles the problem of efficiently computing credal sets over predictions of probabilistic classifiers. To this end, the authors propose perturbing the logits of the model to generate upper and lower bounds of plausible class-wise probabilities while adhering to a likelihood-ratio budget. Additionally, the authors propose an efficient method to compute class-specific credal sets. The proposed method is evaluated against relevant credal prediction baselines on coverage-efficiency trade-off and out-of-distribution detection tasks.

**Strengths:**

The scope of the problem considered - model-agnostic credal predictions - is sizeable and will be of interest to a wide community. Additionally, the post-hoc approach that does not require any retraining as proposed in this paper will encourage its adoption as an added post-training step that can be used to quantify model's epistemic uncertainty. The proposed approach itself, to the best of my knowledge, is sound and the theoretical results seem reasonable, if not unsurprising. The experimental results suggest that the proposed approach is at least on par with the considered credal prediction baselines on the coverage-efficiency trade-off task while allowing, by design, a wide range of coverage. Finally, the efficient computation aspect allows the proposed approach to scale to large models which was previously infeasible.

**Weaknesses:**

Credal predictions are particularly useful in data-scarce and safety-critical domains such as healthcare where the lack of data can lead to higher epistemic uncertainty and understanding the plausible range of model predictions can help avoid catastrophic decisions. In this regard, the motivation behind the need for computationally efficient credal predictions is not very compelling. In the same vein, while the authors note such safety-critical domains in their introduction, the experimental evaluation is primarily on large benchmark image datasets. Lastly, some of the baselines seem to perform slightly better on the OOD detection task, although requiring more computational time.

Overall, I believe the strengths far outweigh the weaknesses.

**Questions:**

Please refer to the weaknesses section. I am mainly unconvinced about the appeal of computational efficiency in so-called *safety-critical* domains where other credal prediction methods, as evident in fig. 3 on the OOD detection task, perform better.

---

> ### Author Response · Authors · 2025-11-20
> **Response to Reviewer XAJU**
>
> We would like to thank you sincerely for taking the time to read our manuscript and provide such valuable feedback. We’re glad that you find our approach interesting to a wide community and appreciate its potential to bring credal prediction to large-scale models, which we fully agree with. We will address the remaining comments and questions below. We have additionally uploaded an updated version of the paper containing requested changes in blue.
>
> **W1:** *“Credal predictions are particularly useful in data-scarce and safety-critical domains such as healthcare where the lack of data can lead to higher epistemic uncertainty and understanding the plausible range of model predictions can help avoid catastrophic decisions. In this regard, the motivation behind the need for computationally efficient credal predictions is not very compelling.”*
>
> Indeed, as you point out, data-scarce scenarios can lead to higher epistemic uncertainty due to the limited knowledge available, which is particularly critical in safety-sensitive decision making, where uncertainty representations such as credal sets are naturally valuable. However, we respectfully disagree with the view that credal predictions are only relevant in data-scarce settings. Even in data-rich contexts, credal sets provide valuable information about prediction reliability and can guide model learning or downstream decision-making. In fact, we show in our work that constructing credal sets for large models like TabPFN and CLIP in data-rich settings improves performance across multiple tasks such as in-context learning (cf. Section 4.3 and 4.4).
>
> In particular, data scarcity is independent of model size, meaning that deploying large models in limited-data settings still requires computationally efficient credal methods. Large architectures such as CLIP and TabPFN are often applied in these scenarios due to their state-of-the-art performance across many tasks. At the same time, in data-scarce settings, these models can exhibit high epistemic uncertainty, which can be effectively captured by credal sets. Thus, our method enhances their practical utility by enabling the construction of such sets for a model like TabPFN which can’t be retrained in practice.
>
> Finally, while determining which settings benefit most from credal predictions is an interesting research question, it is beyond the scope of this work. We are already exploring this direction in related future work, while here we focus on enabling efficient and scalable credal prediction. Having said that, the need for computationally efficient credal predictions is in our opinion well justified across both data-scarce and data-rich scenarios.
>
> **W2:** *“In the same vein, while the authors note such safety-critical domains in their introduction, the experimental evaluation is primarily on large benchmark image datasets."*
>
> We acknowledge that our experiments focus on large benchmark image datasets, which provide a standardized and widely accepted basis for comparison across methods. Our goal in this work is to establish the general effectiveness and scalability of the proposed approach rather than to tailor it to a specific safety-critical domain. The considered benchmarks capture key challenges such as in-context learning, OOD detection, ground-truth coverage and zero-shot classification (cf. Section 4) which transfer directly to safety-critical applications. For example, the natural images used in many of the experiments are common in autonomous robotics applications, such as self-driving cars, where being uncertainty-aware is essential. Having said that, we view domain-specific evaluation as an important next step, having validated the core capabilities of the method on well-established benchmarks.
>
> **W3:** *“Lastly, some of the baselines seem to perform slightly better on the OOD detection task, although requiring more computational time.”*
>
> We agree with you that, while it is true that some baselines show slightly better OOD performance, these methods are computationally infeasible for large models and thus not practical alternatives in such scenarios. Our focus is on developing a method that remains efficient and scalable for large-scale models while maintaining competitive performance across tasks, including OOD detection. If one operates in a setting where models can be (re-)trained and achieving the absolute best OOD performance is the sole priority, other methods may indeed be preferable. However, do note that our OOD experiments primarily aim to demonstrate that our approach can match the performance of state-of-the-art credal methods across diverse tasks, while being orders of magnitude more computationally efficient. Hence, our method makes it feasible to obtain credal predictions for large-scale models, a task that was previously computationally out of reach.

---

### Official Review · Reviewer_yG26 · 2025-11-01

**Soundness:** 3
**Presentation:** 2
**Contribution:** 3
**Rating:** 8
**Confidence:** 3

**Summary:**

This work deals with estimating epistemic uncertainty of predictive models through credal sets. Specifically, it deals with uncertainty estimation for large pretrained foundation models such as LLMs and VLMs, where retraining and finetuning can be exceedingly expensive. The authors address this problem by proposing a training-free method that estimates plausible intervals by modifying the inference procedure. They do so by defining a credal set over the base model’s predictions by adding a variable vector c to their logits; The credal set consists of all the values of this modified predictive distribution that are within a threshold of the maximum likelihood estimate. This credal set allows uncertainty estimation and cautious decision-making. The authors evaluate empirically on 9 domains, including CIFAR, SVHN, FMNIST, DermMNIST, ImageNet, Places365, ChaosNLI, QualityMRI, and TabArena. On these domains, they compare the proposed method with 5 credal uncertainty estimation methods. They evaluate these methods on standard credal classification metrics like coverage and efficiency, on out-of-distribution detection, on active in-context learning, and on zero-shot classification, with the latter two focusing on large pretrained models. They find that the proposed method performs on par with baselines without requiring training and that it provides informative uncertainty estimates for active learning and cautious zero-shot classification.

**Strengths:**

- The proposed method enables uncertainty estimation and cautious inference in large pretrained models like CLIP and TabFPN without expensive retraining & finetuning
- The proposed method is straightforward to implement yet quite effective (as demonstrated by the empirical results)
- The authors introduce credal spider plots to visualize credal sets represented as box intervals

**Weaknesses:**

- While the method does not require retraining, it does require the original training data or an appropriate surrogate to calculate the relative likelihood
- The presentation is unclear in places, e.g., while the credal spider plots are quite informative, a full explanation about what they represent is presented in the appendix, which makes earlier references to them (e.g., figure 1) unclear. Section 4 (Empirical results) presents a lot of information without emphasizing key parts like research questions, datasets, metrics, and baselines; instead interleaves it with details about the setup for each experiment. This section could be made easier to read by explicitly listing the research questions datasets, metrics etc., before the subsections, which may focus on more specific details

**Questions:**

- The proposed method shifts the logits for each class separately, and the conclusion distinguishes this from the more general coupled case. Can you give an example where the class-wise shift would be a bad approximation of the general case?
- Table 4 compares the runtimes of the proposed method and credal baselines, showing that the baselines take 10x more time than the proposed method, but it is not clear how this runtime is defined. Does it include training time or is the 10x difference only in inference time?

---

> ### Author Response · Authors · 2025-11-20
> **Response to Reviewer yG26 (1)**
>
> We thank you for your careful reading of our paper and the valuable, constructive feedback. We are glad that you find our contributions meaningful, specifically, that our method enables credal uncertainty quantification in computationally expensive settings. We will address the remaining comments and questions below. We have additionally uploaded an updated version of the paper containing the requested changes in blue.
>
> **W1:** *“While the method does not require retraining, it does require the original training data or an appropriate surrogate to calculate the relative likelihood”*
>
> Our method can work with the original training data or an appropriate surrogate, but we believe this does not constitute a major drawback. Instead, the ability to use an appropriate surrogate can be seen as a major benefit as we have highlighted in Appendix B. In the case of CLIP, for example, having to use the full dataset (400 million images) would be prohibitive due to the sheer number of training instances. Besides this, further analysis of such scenarios forms an interesting avenue for future research.
>
> **W2:** *“The presentation is unclear in places, e.g., while the credal spider plots are quite informative, a full explanation about what they represent is presented in the appendix, which makes earlier references to them (e.g., figure 1) unclear. Section 4 (Empirical results) presents a lot of information without emphasizing key parts like research questions, datasets, metrics, and baselines; instead interleaves it with details about the setup for each experiment. This section could be made easier to read by explicitly listing the research questions datasets, metrics etc., before the subsections, which may focus on more specific details”*
>
> While an earlier introduction of the credal spider plots would be beneficial, adding it in the main text would disrupt the flow of the paper too much and consume the already limited space. As a compromise, we will, instead, explicitly refer to the existing overview in Appendix C (“Guide on Interpreting Credal Spider Plots”) when showing the credal spider plots for the first time (Figure 1). This should allow the interested reader to quickly locate the relevant explanation.
>
> As for the empirical results section, we highlighted the goals of the experiments in the beginning of Section 4, but we made this now more explicit by rephrasing them into research objectives. Additionally, we added more structure by clearly separating the experimental setup from the results and the interpretation thereof, and slightly rewrote parts of the section. With this, the most important details are mentioned in a structured format in the main text, and the remaining experimental details are deferred to Appendix D, so as not to disrupt the reading flow.

---

> ### Author Response · Authors · 2025-11-20
> **Response to Reviewer yG26 (2)**
>
> **Q1:** *“The proposed method shifts the logits for each class separately, and the conclusion distinguishes this from the more general coupled case. Can you give an example where the class-wise shift would be a bad approximation of the general case?”*
>
> Indeed, allowing joint shifts of all logits results in a strictly more expressive perturbation family than shifting one class at a time. Our class-wise approach can be understood as an inner-approximation of the fully coupled case: by moving along only one logit direction per step, we explore a one-dimensional slice of the simplex based on the model’s original prediction. Figure 1 in the paper gives an example of the probabilities that can be reached this way, illustrated by the, in that case three, paths starting from the MLE prediction. The general case allows movement in any direction in the simplex and can therefore reach any target distribution (within the likelihood budget), because it is not restricted to the one-dimensional slice (one of the three paths in Figure 1). As a consequence, the general case may result in larger probability intervals.
> Most importantly, as our experiments show, this inner approximation yields credal sets with strong coverage and competitive efficiency, while remaining tractable even for large-scale models such as TabPFN and CLIP. Thus, although a fully coupled approach is theoretically more general, the class-wise method offers a trade-off between expressiveness and computational feasibility for practical uncertainty quantification.
>
> **Q2:** *“Table 4 compares the runtimes of the proposed method and credal baselines, showing that the baselines take 10x more time than the proposed method, but it is not clear how this runtime is defined. Does it include training time or is the 10x difference only in inference time?”*
>
> Table 4 reports the values which are plotted in Figure 3 and 8. As mentioned in the main text (e.g., line 332), these numbers refer to training time of each approach. We acknowledge that this is not made explicit in Appendix E.2 and clarified this in the updated version.
> In addition to training time, we have now also compared the inference times of our method and the credal baselines. The results are shown in the table below (also included in the updated Table 4 in the paper). As can be seen, EffCre achieves significantly lower inference time, confirming its computational efficiency during both training and inference.
>
> | Method  | Training time (s)         | Inference time (s)        |
> |---------|----------------------------|----------------------------|
> | EffCre  | 2136.33 ± 1.70             | 1.50 ± 0.02                |
> | CreRL   | 12675.84 ± 412.68          | 11.46 ± 0.12               |
> | CreWra  | 21363.34 ± 33.99           | 11.44 ± 0.19               |
> | CreEns  | 21363.34 ± 33.99           | 11.44 ± 0.23               |
> | CreNet  | 24996.65 ± 180.12          | 11.41 ± 0.18               |
> | CreBNN  | 29796.67 ± 12.94           | 12.74 ± 1.10               |

---

### Official Review · Reviewer_7AnL · 2025-11-14

**Soundness:** 3
**Presentation:** 3
**Contribution:** 3
**Rating:** 6
**Confidence:** 4

**Summary:**

This paper proposes a novel, efficient, and model-agnostic method for credal prediction — a framework for representing epistemic uncertainty via credal sets (i.e., convex sets of probability distributions). The proposed method, termed decalibration, allows the construction of credal predictions without retraining or ensembling, which has historically been a major computational bottleneck in credal learning.

Core Idea
Instead of relying on Bayesian ensembles or multiple retrained models, decalibration works post hoc on a trained classifier. It perturbs the classifier’s logits within a relative-likelihood budget (α), thereby generating class-wise probability intervals that define a credal set.
	•	When α = 1, predictions coincide with the maximum likelihood estimator (MLE).
	•	As α decreases, the set expands, capturing more epistemic uncertainty.

This approach maintains a clear likelihood-based interpretation: the resulting predictions represent all distributions “reachable without sacrificing more than an α-fraction of training likelihood.”

Theoretical Contributions
1.	Convex feasibility and optimization properties:
	•	The set of permissible logit perturbations under a likelihood constraint is shown to be convex and compact on an identifiability hyperplane.
	•	The upper bounds of class probabilities correspond to the solution of a convex optimization problem, while lower bounds are attained at the boundary of this convex region.
2.	Analytical results for 1D (class-specific) logit shifts:
	•	Each class-wise bound can be efficiently computed through small convex programs or simple 1D searches.
	•	The resulting credal sets are nested and monotonic in α.

Empirical Contributions
	•	Extensive experiments show the method’s competitive performance on coverage–efficiency trade-offs and out-of-distribution (OOD) detection, outperforming or matching credal baselines while being orders of magnitude faster.
	•	The approach is scalable to large architectures such as TabPFN and CLIP, where traditional ensemble-based credal methods are computationally infeasible.
	•	Visual tools like credal spider plots are introduced to illustrate uncertainty across multi-class predictions.

Significance
This paper advances epistemic uncertainty quantification by providing a principled yet practical alternative to computationally intensive Bayesian or ensemble-based credal methods. Its theoretical soundness, post-hoc simplicity, and broad applicability to modern foundation models make it a potentially impactful contribution to the ICLR community.

**Strengths:**

•	Originality:
The paper presents a novel post-hoc approach to credal prediction — decalibration — that eliminates the need for retraining or ensemble-based inference, which have been the dominant approaches in credal and epistemic uncertainty estimation. The idea of adjusting logits within a relative-likelihood constraint is both elegant and conceptually original, bridging Bayesian epistemic reasoning with practical optimization.
	•	Technical Quality:
The theoretical exposition is mathematically sound and internally consistent. The authors derive and justify convexity properties of the credal set under the proposed perturbation scheme, ensuring interpretability and computational tractability. Analytical insights into the monotonicity of the α-parameterized likelihood bounds reinforce the approach’s rigor.
	•	Practical Relevance:
The method is computationally efficient and easily applicable to large-scale deep networks and foundation models (e.g., CLIP, TabPFN). This directly addresses a key bottleneck in existing credal learning methods, which often require retraining or expensive ensembles.
	•	Clarity and Presentation:
The paper is generally well written, with clear organization, intuitive explanations, and informative visualizations (e.g., credal spider plots). The connection between likelihood decay and epistemic expansion is articulated clearly and grounded in statistical reasoning.
	•	Experimental Strength:
The experiments are broad and diverse, covering OOD detection, reliability under label noise, and uncertainty calibration across various architectures. The results show consistent improvements in efficiency–coverage trade-offs, supporting the method’s robustness.
	•	Significance:
The proposed framework provides a principled and scalable solution to epistemic uncertainty quantification in deep learning, which is an increasingly critical research area in reliable AI. Its post-hoc and model-agnostic nature make it particularly relevant for the ICLR community focused on trust, calibration, and interpretability.

Overall Strength Summary:
The paper is a solid and meaningful contribution that balances theoretical insight with practical usability. It offers an innovative and efficient solution to credal prediction, addressing both the computational and conceptual limitations of prior approaches.

**Weaknesses:**

•	Limited Theoretical Depth Beyond Convexity:
While the convexity and boundedness of the credal sets are clearly demonstrated, the paper lacks deeper theoretical guarantees. For instance, there are no formal proofs of coverage calibration, robustness under data shift, or asymptotic optimality compared to Bayesian posteriors.
Suggestion: Strengthen the theoretical contribution by connecting decalibration to known uncertainty frameworks such as PAC-Bayesian bounds, conformal coverage guarantees, or distributionally robust optimization.
	•	Potential Overlap with Prior Work:
The approach resembles ideas from temperature scaling, likelihood perturbation, and distributional robustness via logit adjustment (e.g., Stutz et al., 2021; Ahuja et al., 2023). The conceptual novelty might appear incremental to readers unless clearer distinctions are drawn.
Suggestion: Explicitly clarify how decalibration differs mathematically or conceptually from logit perturbation in confidence calibration or adversarial robustness literature.
	•	Empirical Evaluation Scope:
The experimental section, though diverse, is mostly limited to classification tasks. Since credal methods are general, it remains unclear how decalibration performs in structured or regression contexts, where uncertainty has different semantics.
Suggestion: Add at least one structured prediction or regression experiment (e.g., depth estimation or tabular uncertainty).
	•	Interpretability of α-Parameter:
The α hyperparameter controlling likelihood decay is intuitive but empirically opaque. Its practical selection and relationship to epistemic uncertainty remain heuristic.
Suggestion: Provide either a principled selection rule (e.g., based on validation likelihood or calibration metrics) or a sensitivity analysis showing stable performance over α ranges.
	•	Comparative Baselines:
While results are favorable, the baselines do not include recent strong probabilistic calibration models such as Dirichlet Prior Networks or Deep Ensembles with temperature tuning. Without these, the strength of decalibration over modern uncertainty quantifiers remains somewhat uncertain.
Suggestion: Include these baselines or discuss expected trade-offs to contextualize improvements.
	•	Terminological Ambiguity (“Decalibration”):
The term decalibration may be confusing since in standard uncertainty literature, “calibration” typically denotes improving reliability, not relaxing likelihood constraints.
Suggestion: Clarify this choice early in the paper and consider an alternative framing such as “likelihood-scaling credalization” or “post-hoc credal expansion.”
	•	Computational Claims Need Quantitative Backing:
The paper asserts substantial efficiency gains (“orders of magnitude faster”) but provides limited runtime comparisons or profiling details.
Suggestion: Include explicit runtime or FLOPs analysis versus ensemble-based credal methods to substantiate this claim.

Overall Weakness Summary:

The paper is well-executed and conceptually clear, but its mathematical guarantees, empirical breadth, and comparative depth could be strengthened. Clarifying the novelty relative to prior calibration and robustness work, expanding evaluation beyond classification, and providing stronger empirical or theoretical justifications would elevate the paper’s impact and credibility.

**Questions:**

1.	Clarification on the Likelihood Decay Parameter (α):
	•	How should practitioners choose or interpret α in practice?
	•	Is there a connection between α and known uncertainty measures such as expected calibration error (ECE) or Bayesian posterior variance?
	•	Could α be automatically tuned using a validation objective (e.g., coverage vs. set size trade-off)?
2.	Relation to Distributionally Robust Optimization (DRO):
	•	The likelihood-based constraint defining the credal set seems conceptually close to DRO formulations (e.g., χ²-divergence or f-divergence balls).
	•	Can the authors clarify whether decalibration is theoretically equivalent to or inspired by DRO methods?
	•	If not equivalent, how does its uncertainty behavior differ under covariate shift or adversarial perturbations?
3.	Distinction from Prior Post-hoc Calibration Methods:
	•	Decalibration operates directly on logits, similar to temperature scaling, confidence calibration, and logit perturbation techniques.
	•	Could the authors explicitly explain how their formulation mathematically differs from those methods and why it better captures epistemic (not aleatoric) uncertainty?
4.	Computational Complexity Claims:
	•	The paper claims “orders of magnitude” improvement in efficiency over ensemble-based methods.
	•	Could the authors provide quantitative runtime comparisons (e.g., seconds per image or FLOPs) for fair assessment?
	•	Is the optimization step fully parallelizable, and how does performance scale with the number of classes?
5.	Generalization to Regression or Structured Outputs:
	•	The current framework appears classification-specific.
	•	Is there a theoretical extension of decalibration to continuous outputs (e.g., regression) or structured prediction (e.g., segmentation, detection)?
	•	If so, how would the likelihood constraints translate?
6.	Credal Set Visualization and Intuition:
	•	The “credal spider plots” are compelling but may lack clear interpretability for practitioners.
	•	Could the authors provide an example of how such visualization could inform human decision-making (e.g., in safety-critical applications)?
7.	Uncertainty Decomposition:
	•	Does decalibration allow separating epistemic vs. aleatoric uncertainty components?
	•	If not, could an ensemble of decalibrated models or Bayesian prior over α achieve that?
8.	Robustness under Data Shift:
	•	Have the authors tested decalibration under distributional shift scenarios (e.g., corrupted datasets, domain transfer)?
	•	If so, how does it compare to ensemble or conformal methods in terms of coverage stability?

Summary of Key Questions for Rebuttal Focus:
	1.	Theoretical connection to DRO and uncertainty calibration frameworks.
	2.	Justification and interpretation of α.
	3.	Explicit differentiation from existing logit-perturbation and calibration methods.
	4.	Quantitative validation of efficiency claims.
	5.	Potential extension beyond classification tasks.

**Details Of Ethics Concerns:**

This paper focuses on theoretical and algorithmic advances in credal prediction and epistemic uncertainty quantification, with no experiments involving human subjects, private data, or potentially harmful applications.

The datasets used (e.g., standard vision and tabular benchmarks) are public and widely accepted in the machine learning community, and there is no indication of ethical risk such as bias amplification, privacy violation, or misuse potential.

The methodology enhances model transparency and uncertainty communication, which—if anything—supports ethical AI development rather than compromising it.

---

> ### Author Response · Authors · 2025-11-20
> **Response to Reviewer 7AnL (1)**
>
> We thank you for reading our work and for providing such extensive feedback. We appreciate your recognition that our approach advances epistemic uncertainty representation by offering a principled yet practical alternative to computationally expensive Bayesian and credal prediction methods. We address your remaining comments and questions below. We have additionally uploaded an updated version of the paper containing requested changes in blue.
>
> **W1 & Q2:** *"Limited Theoretical Depth Beyond Convexity: While the convexity and boundedness of the credal sets are clearly demonstrated, the paper lacks deeper theoretical guarantees. For instance, there are no formal proofs of coverage calibration, robustness under data shift, or asymptotic optimality compared to Bayesian posteriors. Suggestion: Strengthen the theoretical contribution by connecting decalibration to known uncertainty frameworks such as PAC-Bayesian bounds, conformal coverage guarantees, or distributionally robust optimization."*
>
> Thank you for noting that our paper already provides theoretical contributions, including the convexity and boundedness of the credal sets. We agree that extending the theory to include formal coverage guarantees, robustness under data shift, or connections to PAC-Bayesian, conformal, or distributionally robust frameworks is an exciting direction.
>
> However, these extensions go beyond the scope of the current work. Our primary goal here is to introduce decalibration as a practical, principled approach for generating credal sets from a single trained model, supported by core theoretical properties and extensive empirical evaluation. We see the broader theoretical unification you suggest as an excellent and natural direction for future research.
>
> **W2, Q5 & Q8:** *"Empirical Evaluation Scope: The experimental section, though diverse, is mostly limited to classification tasks. Since credal methods are general, it remains unclear how decalibration performs in structured or regression contexts, where uncertainty has different semantics. Suggestion: Add at least one structured prediction or regression experiment (e.g., depth estimation or tabular uncertainty)."*
>
> We appreciate your acknowledgement that our experiments are diverse and provide a solid empirical evaluation. While credal sets are indeed a general concept [1,2], in practice they are predominantly applied in machine learning in classification scenarios (see [3-7]).
>
> We agree that extending decalibration to regression or structured prediction tasks, such as depth estimation or tabular uncertainty, is both interesting and valuable from an uncertainty perspective. However, such extensions are theoretically nontrivial and would require careful development beyond the scope of the current work. That said, we do recognize the importance of tabular uncertainty, which is why we already explore it in the context of active in-context learning using TabPFN.
>
> [1] Peter Walley. Statistical reasoning with imprecise probabilities. 1991.
>
> [2] Isaac Levi. On indeterminate probabilities. 1978.
>
> [3] Wang et al. Credal wrapper of model averaging for uncertainty estimation on out-of-distribution detection. 2024.
>
> [4] Wang et al. Credal deep ensembles for uncertainty quantification. 2024.
>
> [5] Nguyen et al. Credal ensembling in multi-class classification. 2025.
>
> [6] Caprio et al. Imprecise bayesian neural networks. 2023.
>
> [7] Antonucci et al. Likelihood-based robust classification with bayesian networks. 2012.
>
> [8] Abellán et al. Disaggregated total uncertainty measure for credal sets. 2006.

---

> ### Author Response · Authors · 2025-11-20
> **Response to Reviewer 7AnL (2)**
>
> **W3 & Q1:** *"Interpretability of *$\alpha$* -Parameter: The *$\alpha$*  hyperparameter controlling likelihood decay is intuitive but empirically opaque. Its practical selection and relationship to epistemic uncertainty remain heuristic. Suggestion: Provide either a principled selection rule (e.g., based on validation likelihood or calibration metrics) or a sensitivity analysis showing stable performance over *$\alpha$* ranges."*
>
> We want to clarify that coverage and efficiency inherently involve a trade-off: improving one generally comes at the expense of the other. The $\alpha$ hyperparameter controls the likelihood constraint and allows one to navigate this trade-off: lower $\alpha$ favors coverage, while higher $\alpha$ favors efficiency. Its relationship to epistemic uncertainty can be seen as follows: $\alpha$ allows one to express one’s belief about the quality of the maximum likelihood estimate (MLE) that can be obtained from the available (training) data sample. If the MLE is believed to be close to the ground-truth model, in terms of their predicted distributions, a large $\alpha$ can be chosen, resulting in a smaller set (reflecting confidence that the ground-truth lies close to the MLE), and vice versa.
> Practically, the value of $\alpha$ may be determined by evaluating the coverage and efficiency on a validation set.
> Importantly, our experiments show that EffCre achieves strong performance across this trade-off based on different values of $\alpha$ compared to state-of-the-art baselines. In addition, we provide an ablation study on $\alpha$ for active in-context learning in Appendix F.2 (Fig. 14). Finally, we consider this adaptability to be a key advantage of our approach: one can adjust $\alpha$ to suit the task at hand.
>
> **W4:** *"Terminological Ambiguity (“Decalibration”): The term decalibration may be confusing since in standard uncertainty literature, “calibration” typically denotes improving reliability, not relaxing likelihood constraints. Suggestion: Clarify this choice early in the paper and consider an alternative framing such as “likelihood-scaling credalization” or “post-hoc credal expansion.”*
>
> The term decalibration is introduced and motivated directly in the introduction:
> > Building on a likelihood-based notion of plausibility, we construct credal predictions from a single trained model by decalibration: we systematically perturb the model’s logits so that the resulting probabilities move away from the maximum-likelihood fit while staying within a prescribed relative-likelihood budget.
>
> We also provide intuition for the terminology: calibration adjusts probabilities to be more correct, whereas decalibration explores how much they can be changed while still remaining supported by the data. Moreover, we explicitly frame our approach in the introduction and throughout the paper as a post-hoc method for credal prediction, which aligns with your suggested phrasing of “post-hoc credal expansion”. Given these explanations and framing choices, we believe the concept and terminology of decalibration are clearly introduced and justified in the present manuscript.
>
> **W5 & Q4:** *"Computational Claims Need Quantitative Backing: The paper asserts substantial efficiency gains (“orders of magnitude faster”) but provides limited runtime comparisons or profiling details. Suggestion: Include explicit runtime or FLOPs analysis versus ensemble-based credal methods to substantiate this claim."*
>
> Our method indeed provides substantial efficiency gains, which can be seen from multiple perspectives:
>
> 1. Number of model trainings required: Our approach requires zero additional model training, as it can be applied on top of any probabilistic classifier. Ensemble-based credal methods [3-7], by contrast, need to train multiple models. For example, a 10-member ensemble (Fig. 13) requires 9 additional models, naturally resulting in much higher compute compared to our method (Fig. 3).
>
> 2. Measured runtime: We report training and inference times under identical computational settings in Table 4 and Fig. 3. These results confirm that our method is orders of magnitude faster than state-of-the-art baselines.
>
> 3. Large-scale pretrained models: TabPFN was trained for 20 h on 8 RTX 2080 Ti GPUs, and CLIP for 12 days on 256 NVIDIA V100 GPUs [9, 10]. Our method applies directly to these models at zero additional cost, while credal baselines would require retraining each ensemble member (e.g. 10), making them practically infeasible.
>
> Having said that, we believe that a more detailed runtime analysis does not change or strengthen the already well-supported conclusion: our method is computationally far more efficient than ensemble-based credal approaches.
>
> [9] Hollmann et al. TabPFN: A transformer that solves small tabular classification problems in a second. 2023.
>
> [10] Radford et al. Learning Transferable Visual Models From Natural Language Supervision. 2021.

---

> ### Author Response · Authors · 2025-11-20
> **Response to Reviewer 7AnL (3)**
>
> **Q6:** *"Credal Set Visualization and Intuition: • The “credal spider plots” are compelling but may lack clear interpretability for practitioners. • Could the authors provide an example of how such visualization could inform human decision-making (e.g., in safety-critical applications)?"*
>
> A detailed “Guide on Interpreting Credal Spider Plots” is provided in Appendix C. We added this guide specifically to improve the interpretability of the visualizations for practitioners, addressing the potential difficulty of reading the plots at a glance. The guide explains how to interpret results and extract information about the represented uncertainty. For example, in safety-critical scenarios, the plots can highlight high-uncertainty predictions, helping users identify potential risks and make informed decisions.
>
> **Q7:** *"Uncertainty Decomposition: • Does decalibration allow separating epistemic vs. aleatoric uncertainty components? • If not, could an ensemble of decalibrated models or Bayesian prior over α achieve that?"*
>
> Our decalibration method generates credals sets which are not constrained to any particular decomposition of credal uncertainty. We use the axiomatically justified approach proposed by Abellan et al. [8], which provides widely used measures to separate aleatoric and epistemic uncertainty that can be directly applied to the credal sets generated by decalibration. Still, other uncertainty measures may also be adopted depending on the specific requirements of the application of interest.
>
> [8] Abellán et al. Disaggregated total uncertainty measure for credal sets. 2006.

---

### Author Response · Authors · 2025-11-20
**General Response**

**We thank all reviewers for their thoughtful comments, constructive feedback, and general appreciation of our work.** We answered the comments and questions in responses to the individual reviewers, and carefully revised the manuscript in response to the suggestions. All changes in the paper are marked in $\textcolor{blue}{blue}$. Below is a list of all changes:


- Added a sentence in the Abstract to clearly separate the empirical claims.
- Added an explicit reference in the caption of Figure 1 to the guide on interpreting spider plots in the appendix.
- Revised beginning of Section 4 (Empirical Results) to more clearly connect the research objectives to the corresponding experiments.
- Restructured the evaluation so that each experiment now has a distinct experimental setup and results paragraph.
- Added descriptions and performance of two additional baselines (Evidential Deep Learning and Deep Deterministic Uncertainty) in Appendix D.3 and E.2.
- Added evaluation of inference time, accuracy and expected calibration error in Appendix E.2 and F.3.

---

### Author Response · Authors · 2025-12-03
**Summary of the Discussion**

Dear AC, dear reviewers,

We thank all reviewers again for the valuable feedback they provided, which has improved our work. All the changes that were discussed have been incorporated in the paper and can be found in the latest revision. As we have not received any further response from the reviewers, we believe the remaining comments have been resolved. Below, we briefly summarize the main points of the discussion with the reviewers.

**Reviewer 7AnL**

We highlighted the formal analysis of our method in the paper, discussed the choice of $\alpha$ in our approach, and provided a more detailed runtime analysis.

**Reviewer yG26**

Initiated by the reviewer’s comments, we further improved the readability of the empirical section by putting a stronger emphasis on the research objectives and its connection to the experiments, and by providing more structure to separate the experimental details from the concrete results.

**Reviewer XAJU**

The reviewer recognized that our work is of interest to a broad community and our discussion raised interesting next steps that may lead to valuable future work.

**Reviewer 9AwT**

In response to the reviewer’s main comment, we added further comparisons to evidential deep learning and deterministic baselines. This strengthened our contribution by demonstrating that our method performs competitively beyond the credal predictors we focused on in our submission. The reviewer acknowledged the additional results and intended to raise their score accordingly.
In addition, we included results comparing the inference time of our method with the baselines, further emphasizing that our approach is substantially more efficient than the competing methods.

---

### Meta-Review · Area_Chair_Fosu · 2026-01-12

**Summary:**

Based on reviewer discussions, all reviewers have indicated a willingness to accept the paper. I personally have some reservations about the paper's discussion of coverage: The paper defines coverage based on the ground-truth conditional. In classification, this is never observed! Only a _sample_ from this distribution (i.e. a label) is observed. So how can this be calculated?

Nevertheless, the paper contains many other demonstrations, including active learning, and OoD detection.

**Reviewer Concerns:**

Particularly important was the resolution of the importance of alpha, and the results on training/inference time to support the main claim of efficiency.

**Reviewer Scores:**

Most reviewers engaged well with the discussion, particularly 9AwT indicated direct willingness to increase score.

---

### Decision · Program_Chairs · 2026-01-26

Accept (Poster)